# PRICIN: PRINCIPLE-CENTERED INORGANIC RETROSYNTHESIS

## ABSTRACT

Bridging the gap between what is designable by computational discovery and what is synthesizable in the lab remains a central obstacle for closed-loop materials science. We tackle single-step inorganic retrosynthesis and show that explicit chemical principles are potent inductive biases for learning to plan syntheses. We introduce **PRICIN**, a principle-centered approach that reformulates precursor planning around two laws: elemental conservation and electron balance. PRICIN embeds stoichiometry and oxidation-state semantics directly into the target representation via two pretraining objectives, including an auxiliary oxidation-state supervision that injects charge awareness. At inference, a lightweight element-wise filter first predicts the required number of precursors and then prunes candidates that violate conservation constraints, yielding explainable, chemically consistent precursor sets without external retrieval or rigid templates. Across the Retrieval-Retro (year-split) and Ceder benchmarks, PRICIN attains state-of-the-art performance on Top-$k$ and combination Top-$k$ metrics, improving over the previous best by **+5.17** Top-1 and by up to **+20.78** percentages on Top-20. Ablations confirm that oxidation-state supervision and conservation-aware filtering are both necessary and complementary, substantially reducing early-rank errors. The code will be released upon acceptance.

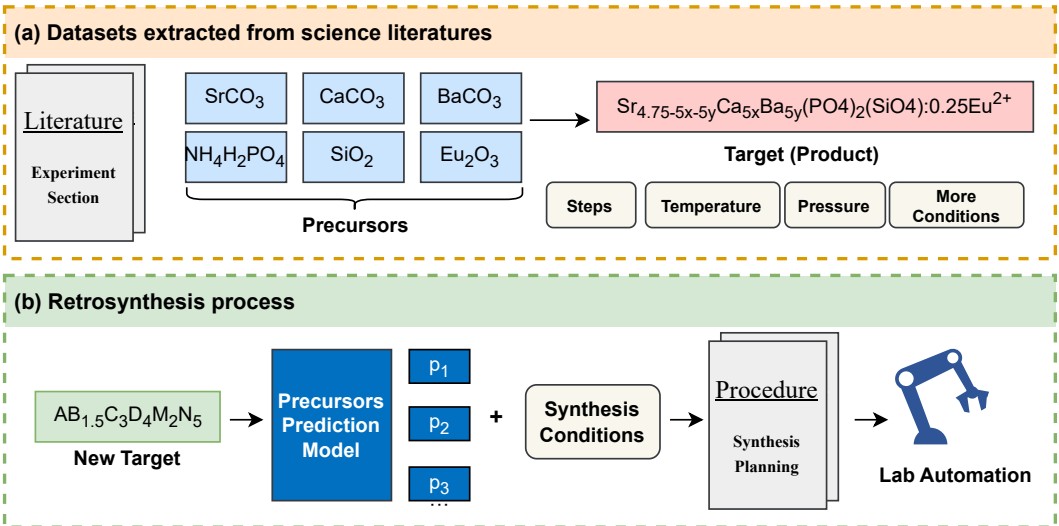

Figure 1: **The inorganic retrosynthesis task.** (a) Prior work has focused on extracting synthesis recipes from scientific literature, where the precursors and conditions for a known target are documented. (b) In contrast, the retrosynthesis process inverts this problem: given a novel target compound, the goal is to predict a set of viable precursors and synthesis conditions. This automated synthesis planning is a key step toward the ultimate vision of using lab automation to accelerate the discovery and synthesis of new materials.

## 1 INTRODUCTION

The core objective of materials science is to discover and deploy new materials with superior properties for critical applications, including semiconductors (Choubisa et al., 2023), energy storage (Yao et al., 2023) and bio-materials (McDonald et al., 2023). While traditional materials exploration relies heavily on trial and error, the advent of theoretical and first-principles calculations has significantly accelerated the design phase. Systems like GNoME (Merchant et al., 2023) and MatterGen (Zeni et al., 2023) can now efficiently generate promising compositions and structures from vast candidate spaces. However, with the rise of the closed-loop paradigms such as A-Lab (Szymanski et al., 2023b), the primary bottleneck in the full Design-Make-Analyze-Test (DMAT) cycle has become increasingly apparent: many existing models excel at screening and design but struggle to provide actionable synthesis routes, creating a significant gap between what is designable and what is synthesizable.

Similar to organic chemistry, the synthesis of inorganic materials can be framed as a retrosynthesis problem. However, the two fields have significant differences that make direct translation of methods unfeasible: (1) the lack of large-scale, standardized datasets comparable to the USPTO database (Somnath et al., 2021); (2) the greater difficulty in calculating the properties and structures of inorganic crystals (Ratcliff et al., 2017), which involve larger numbers of atoms and periodic boundary conditions; (3) the critical role of stoichiometry in determining both the target crystal structure and the feasible precursor combinations; and (4) the absence of universally transferable reaction centers (Lan et al., 2024) or transition state mechanisms (Zhang et al., 2016; Ucak et al., 2022) in inorganic solid-state synthesis, making data-driven modeling more challenging. Consequently, inorganic retrosynthesis has emerged as a key unsolved problem for achieving closed-loop materials discovery.

In recent years, automated methods for inorganic retrosynthesis have begun to emerge. Thermodynamics-based pathfinding approaches frame solid-state synthesis as a search for a feasible path on a reaction network, where edge weights encode proxies for reaction energy and phase competition, successfully reproducing literature routes and proposing candidates for new targets (McDermott et al., 2021; Miura et al., 2021). For precursor set generation, early work used text mining and generative models to learn synthesis planning from literature (Kim et al., 2020). More recent methods like SynthesisSimilarity (He et al., 2023) learn material similarity from historical recipes to recommend precursors by analogy, ElementwiseRetro (Kim et al., 2022) uses a template-based GNN ranker, and RetrievalRetro (Noh et al., 2024) incorporates a reaction energy-based retriever. While these methods have advanced the field, they primarily rely on precedent-based learning, with limited explicit modeling of chemical principles. This can lead to early-ranking errors, inflexibility in handling multi-source precursors.

In this work, we propose a principle-centered approach (PRICIN) to inorganic retrosynthesis. We reformulate the precursor planning task around two core chemical reaction laws: **elemental conservation** and **electron balance** (the total increase in oxidation states equals the total decrease). Instead of relying solely on templates or retrieval, we design two tasks that embed elemental stoichiometry and oxidation-state directly into the target material's representation. One objective uses auxiliary supervision on oxidation states to implicitly model valence changes during reaction, while the other models compositional ratios to ensure elemental conservation. We also introduce a lightweight element-wise filter that first predicts the required number of precursors and then filters out candidates that violate elemental conservation, ensuring that elements of every precursor are sourced validly. Guided by these principles, PRICIN generates explainable and synthesizable precursor sets without relying on external retrieval and significantly increases top-$k$ accuracy.

We systematically evaluate our method on the Retrieval-Retro (Noh et al., 2024) and Ceder (Kononova et al., 2019) benchmark datasets, with the former using a year-based split to test for temporal generalization. Our results demonstrate state-of-the-art performance across multiple metrics, validating the effectiveness and practical potential of a principle-centered approach to inorganic retrosynthesis. On the Retrieval-Retro dataset, PRICIN achieves 66.19% Top-1 and 86.52% Top-20 combination accuracy, outperforming the previous best method (Retrieval-Retro) by +5.17% and +17.29% respectively. Similarly, on the Ceder dataset, PRICIN achieves 61.96% Top-1 and 81.24% Top-20 combination accuracy, improving over the previous best by +5.18% and +14.13% respectively.

Our contributions are summarized as follows:

Table 1: **Capability comparison of retrosynthesis methods.** Our approach is benchmarked against prior works, highlighting key features for successful precursor prediction: the ability to retrieve from a database of known reactions, the integration of explicit chemical domain knowledge, and the capacity to extrapolate predictions to novel materials.

| Model | Retrieval capability | Chemical domain knowledge | Extrapolation to new systems |
|---|---|---|---|
| ElemwiseRetro (Kim et al., 2022) | ✗ | Low | Medium |
| Synthesis Similarity (He et al., 2023) | ✓ | Low | Low |
| Retrieval-Retro (Noh et al., 2024) | ✓ | Medium | Medium |
| Retro-Rank-In (Prein et al., 2025) | ✗ | Low | High |
| **Ours** | ✓ | High | High |

- **Explicit Oxidation-State Supervision.** To the best of our knowledge, we are the first to propose that explicit oxidation-state supervision should be a core component of modeling inorganic retrosynthesis, providing a direct chemical signal that significantly improves prediction accuracy.

- **Principle-Centered Formulation.** We develop a comprehensive modeling framework that reconstructs the retrosynthesis task around two key chemical principles: elemental conservation and electron balance, integrating these rules into both the learning process and inference constraints.

- **Effective Element-Wise Filter.** We employ a simple and highly efficient element-wise filter at inference time that first predicts the number of precursors and then prunes illegal candidates, leading to consistent and substantial accuracy improvements with minimal computational overhead.

## 2 RELATED WORK

**Literature mining and datasets.** Recent literature-mining efforts have demonstrated that large-scale, automated extraction of inorganic synthesis knowledge is feasible and directly enables recipe-level datasets. Kononova et al. (2019) builds a large-scale, automatically text-mined corpus of solid-state synthesis "recipes" by scraping articles, detecting synthesis paragraphs and converting text into structured JSON records that capture targets, precursors, operations. Huo et al. (2019) uses semi-supervised topic modeling to discover interpretable step-topics and classifies synthesis modalities while reconstructing procedural order. He et al. (2020) develops a two-step pipeline that masks material mentions and infers roles with a BiLSTM-CRF, assembling a large corpus of precursors and targets and proposing a precursor-similarity metric that supports reactant substitution. Wang et al. (2022b) extends mining to solution-based syntheses with a publisher-scale pipeline combining a BERT-based paragraph classifier. Wang et al. (2022a) introduces a unified ontology (ULSA) and a learned mapping from text to standardized action graphs, providing a common procedural vocabulary, allowing operation prediction and full-step synthesis planning.

**Precursor recommendation.** McDermott et al. (2021) cast solid-state synthesis planning as pathfinding on a thermochemistry-derived reaction network whose edge weights encode thermodynamic proxies, recovering literature routes (e.g., $YMnO_3$, $Y_2Mn_2O_7$, $Fe_2SiS_4$, $YBa_2Cu_3O_{6.5}$) and proposing routes to unseen targets. Miura et al. (2021) recast ceramic synthesis as a sequence of pairwise interfacial reactions, rank interface reactivity via ab-initio thermodynamics, and predict the earliest nonequilibrium intermediates that steer phase evolution. Kim et al. (2020) mine the materials-science literature with an NLP pipeline (ELMo/FastText embeddings and NER) and train an unsupervised conditional VAE to model synthesis actions and precursors conditioned on a target compound, retrospectively proposing plausible precursors for unseen perovskites such as $InWO_3$ and $PbMoO_3$ while providing literature-trained representations that complement thermodynamic checks. He et al. (2023) propose a data-driven strategy (SynthesisSimilarity) that learns a neural notion of chemical similarity from 29,900 literature recipes to recommend precursor sets for novel targets by analogy to historically synthesized materials. Kim et al. (2022) develop a graph neural framework (ElementwiseRetro) that ranks precursor sets under a probabilistic template requiring each target element be sourced from exactly one precursor, a constraint that can limit flexibility for multi-source routes. Noh et al. (2024) introduce a retrieval-based approach (RetrievalRetro) that

implicitly extracts precursor information from reference materials and injects thermodynamic priors via a Neural Reaction Energy retriever, yet can propagate early ranking errors to higher-$k$ lists.

**Applications and LLMs.** Chen et al. (2024) formalizes a thermodynamic strategy for solid-state synthesis and validates these principles robotically across hundreds of reactions with routinely higher phase purity than traditional recipes. Song et al. (2025) introduces CSLLM, a domain-adapted LLM that predicts synthesizability, methods, and precursors from a crystal-text representation and a large curated corpus, though without explicitly enforcing charge balance. Several other domain-specialized language models have been developed for materials science, such as MatBERT (Walker et al., 2021) and MatSciBERT (Gupta et al., 2022). Szymanski et al. (2023a) presents A-Lab, a closed-loop platform that fuses ab initio phase-stability priors, literature-trained recipe models, robotics, ML-based XRD, and active learning to reliably realize computationally proposed oxides and phosphates.

## 3 PRELIMINARY

We consider the single–step inorganic retrosynthesis problem: given a target compound $x$ and synthesis conditions $(\mathcal{T}, \mathcal{P})$, predict a multiset of precursors $C = \{p_i\}_{i=1}^m$ together with stoichiometric coefficients $\mathbf{s} = (s_1, \ldots, s_m)$ and optional byproducts $B = \{b_j\}_{j=1}^r$ with coefficients $\mathbf{t} = (t_1, \ldots, t_r)$ such that the reaction is chemically consistent and thermodynamically driven.

**Elemental conservation.** Let $\mathcal{E}$ denote the set of all chemical elements under consideration, with $N_E = 118$ denoting the total number of elements considered, and let $E \in \mathcal{E}$. Let $n(E, y)$ denote the count of element $E$ in material $y$ per formula unit. Element balance requires that there exist non–negative integers $\{s_x\}, \{s_i\}$ and $\{t_j\}$ satisfying, for all $E$ in the element set $\mathcal{E}$,

$$\sum_{i=1}^m s_i \, n(E, p_i) = s_x \, n(E, x) + \sum_{j=1}^r t_j \, n(E, b_j). \tag{1}$$

In practice, when multiple precursor sets may synthesize the same target, we relax per-atom balance to the following element-level formulation: We write $M(y)$ for "the set of elements that appear in material $y$." The coverage constraint is simply

$$\bigcup_{i=1}^m M(p_i) = M(x) \cup \bigcup_{j=1}^r M(b_j). \tag{2}$$

**Electron balance.** Let $z(E, \cdot)$ denote admissible oxidation states. Chemical principles dictate that each compound must be charge–neutral and that the overall reaction must be redox-balanced. To formalize charge neutrality, we account for elements with multiple oxidation states by defining the net charge of a compound $y$ as a sum over its constituent species $\sigma$ (element-oxidation pairs): $Q(y) = \sum_\sigma z(\sigma, y) \, n(\sigma, y)$, where $z(\sigma, y)$ is the charge of species $\sigma$ and $n(\sigma, y)$ is its count. Consequently, for any valid compound: $Q(x) = 0, Q(p_i) = 0, Q(b_j) = 0$. Beyond the neutrality of individual compounds, a valid chemical reaction must also maintain redox balance. This principle requires that the sum of oxidation state changes across all elements in the reaction be zero, meaning any oxidation is precisely balanced by a reduction. Therefore, the valence states of the precursors are fundamentally linked to those of the target material, motivating our subsequent use of precursor oxidation states as an informative signal for our model.

**Problem statement.** Inorganic retrosynthesis seeks to find and rank precursor sets $(C, \mathbf{s})$ (and optional $B, \mathbf{t}$) that satisfy elemental conservation and charge consistency. Downstream sections instantiate this definition with learnable embeddings, retrieval evidence, and decision–time constraint checks.

## 4 METHODS

**Overview.** Our proposed pipeline, illustrated in Figure 2, consists of a multitask training stage followed by a constrained ranking stage. Given a target compound $x$, we first apply a fixed feature extraction module $h(\cdot)$ to obtain its compositional representation, $\mathbf{x}_{\text{feat}} = h(x)$. This feature vector is

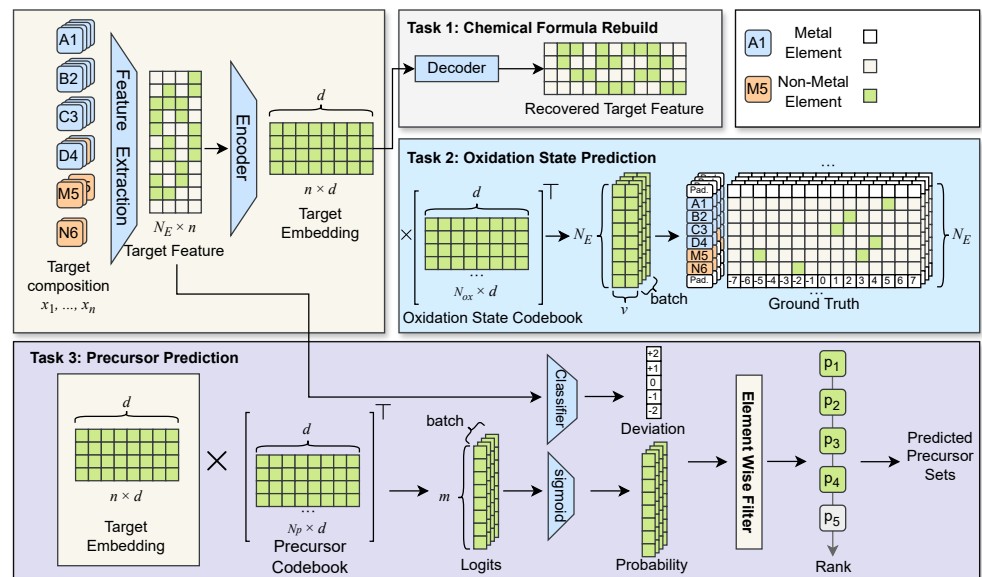

Figure 2: **Overview of our chemical principle-centered pipeline.** The model takes a target composition, extracts features, and encodes it into a fixed-dimensional target embedding. This embedding is trained via three auxiliary tasks: (1) **Chemical Formula Rebuild**, where a decoder reconstructs the elemental fractions of the target to preserve stoichiometric information; (2) **Oxidation State Prediction**, where the model learns to predict the distribution of oxidation states for each element by comparing the target embedding to a species codebook, thereby injecting oxidation-awareness; and (3) **Precursor Prediction**, where the target embedding is used to rank candidates from a precursor codebook. In the final step, an **Element-wise Filter** is applied to the ranked precursor probabilities to enforce elemental conservation, yielding a list of chemically coherent precursor sets.

then passed to a trainable encoder $f_\theta(\cdot)$ to produce a chemically-aware target embedding in the latent space, $\mathbf{t} = f_\theta(\mathbf{x}_{\text{feat}})$. This embedding is jointly optimized against three objectives using two distinct learnable codebooks: an **Oxidation State Codebook**, $\mathcal{C}_{\text{ox}} \in \mathbb{R}^{N_{ox} \times d}$ ($N_{ox} = v \times N_E$, $v$ stands for all possible 15 oxidation states for an element), and a **Precursor Codebook**, $\mathcal{C}_{\text{prec}} \in \mathbb{R}^{N_p \times d}$. The objectives are: reconstructing the initial compositional features, predicting oxidation states, and predicting the precursor set. Finally, the ranking stage uses the optimized target embedding to generate precursor recommendations, which are refined by a element wise filter that enforces elemental conservation.

## 4.1 TASK 1: CHEMICAL FORMULA REBUILD.

The goal of this task is to train the encoder $f_\theta$ to produce a robust target embedding $\mathbf{t}$ that retains the essential stoichiometric information of the original compound. To achieve this, we employ a decoder, $d_\phi(\cdot)$, which attempts to reconstruct the initial target feature vector from the embedding, i.e., $\hat{\mathbf{x}}_{\text{feat}} = d_\phi(\mathbf{t})$. This autoencoding structure ensures that the learned embedding is a compressed but faithful representation of the compound's composition. The reconstruction loss is formulated as a binary cross-entropy between the original and reconstructed feature vectors:

$$\mathcal{L}_{\text{frac}} = -\sum_{i=1}^{|\mathcal{E}|} \left[ x_{\text{feat},i} \log(\hat{x}_{\text{feat},i}) + (1 - x_{\text{feat},i}) \log(1 - \hat{x}_{\text{feat},i}) \right], \tag{3}$$

where $\mathbf{x}_{\text{feat}}$ is the ground-truth feature vector and $\hat{\mathbf{x}}_{\text{feat}}$ is the reconstructed vector. This task serves as a regularization objective that enhances the stability and generalization of downstream tasks.

## 4.2 TASK 2: OXIDATION STATE PREDICTION.

Chemical reactions are governed by charge conservation, but the oxidation states in a novel target compound are often unknown (e.g., $Ba_{0.5}Sr_{0.5}Co_xFe_{1-x}O_{3-\delta}$). We address this by predicting the target's oxidation state distribution, using the known states of its precursors as the ground truth. Specifically, we use the learned target embedding $\mathbf{t}$ to query a learnable **Oxidation State Codebook** $\mathcal{C}_{ox}$. This codebook stores an embedding for each possible element-oxidation state pair. We compute a logit for each pair via a matrix-vector product: $\mathbf{z}_{ox} = \mathcal{C}_{ox}^{\top}\mathbf{t}$. The logits are then passed through a sigmoid function to yield the predicted probability distribution, $\hat{\boldsymbol{\pi}} = \sigma(\mathbf{z}_{ox})$. The model is trained to match this prediction to the ground-truth distribution $\boldsymbol{\pi}^{\star}$, which is derived from the aggregated oxidation states of the precursors. The loss is:

$$\mathcal{L}_{ox} = -\sum_E \sum_z \left[ \pi_{E,z}^{\star} \log(\hat{\pi}_{E,z}) + (1 - \pi_{E,z}^{\star}) \log(1 - \hat{\pi}_{E,z}) \right]. \tag{4}$$

This task injects charge awareness into the target embedding, encouraging a latent-space representation that reflects valence chemistry.

## 4.3 TASK 3: PRECURSOR PREDICTION

**Limitations of implicit count prediction.** A fundamental challenge in retrosynthesis is determining the number of precursors, $m$, required for a target material $x$. Prior methods often address this implicitly, for instance, by applying a fixed threshold (e.g., 0.5) to a sigmoid output layer to select a precursor set. This approach is suboptimal: if the number of precursors selected via thresholding does not match the ground truth $m$, the model is trained on an incorrect premise, forcing it to merely redistribute probabilities over an erroneously sized set rather than correcting the count itself.

**Explicit count prediction as classification.** A more principled approach is to first predict the number of precursors $m$, and subsequently predict their identities. As illustrated in Figure 2 (Task 3), we introduce a dedicated precursor count predictor. This module frames the task as a classification problem, guided by the chemical heuristic that $m$ often correlates with the number of non-metallic elements, $E_{non}$, in the target (e.g., O, N, F, Cl, Br). Instead of predicting $m$ directly, the classifier learns to predict the deviation of $m$ from $E_{non}$. Specifically, it outputs a value from the discrete set $\{-2, -1, 0, +1, +2\}$, which represents the predicted difference $m - E_{non}$. This transforms the prediction of an arbitrary integer into a constrained, chemically-informed classification problem, allowing the model to learn a robust prior over the size of the precursor set.

**Precursor prediction Loss.** We use the target embedding $\mathbf{t} = f_\theta(\mathbf{x}_{feat})$ to predict the precursor set. We compute a logit vector $\mathbf{z} \in \mathbb{R}^{N_p}$ by taking a matrix-vector product between the target embedding and the **Precursor Codebook**, $\mathbf{z}_{prec} = \mathcal{C}_{prec}^{\top}\mathbf{t}$. The selection probabilities for all $N_p$ library precursors are then obtained by an elementwise sigmoid function, $\hat{\mathbf{y}} = \sigma(\mathbf{z}_{prec})$. Training uses a multi-label binary cross-entropy over the full library:

$$\mathcal{L}_{prec} = -\sum_{i=1}^{N_p} \left[ y_i \log(\hat{y}_i) + (1 - y_i) \log(1 - \hat{y}_i) \right], \tag{5}$$

where $y_i = 1$ if the $i$-th precursor is in the ground-truth set $C$, and $y_i = 0$ otherwise.

**Total objective.** The final training loss is a weighted sum:

$$\mathcal{L} = \lambda_{frac}\mathcal{L}_{frac} + \lambda_{ox}\mathcal{L}_{ox} + \lambda_{prec}\mathcal{L}_{prec}. \tag{6}$$

## 4.4 ELEMENT-WISE FILTER

Existing methods implicitly learn precursor selection over the entire candidate set. Although this approach achieves reasonable prediction performance, the top-$k$ ranked candidates may still contain **chemically invalid choices** that violate elemental conservation principles. To address this limitation, we introduce an **Element-Wise Filter** as a post-processing step during inference. As established in Section 3, a valid precursor must satisfy the elemental conservation constraint, i.e., its non-volatile

elements should form a subset of the target's elements. However, solid-state synthesis typically proceeds at elevated temperatures (often exceeding 800°C), under which certain elements—specifically carbon (C), hydrogen (H), and nitrogen (N)—undergo thermal decomposition and are released as gaseous byproducts ($CO_2$, $H_2O$, $NO_x$/$NH_3$). Consequently, precursors containing these elements can legitimately contribute to targets that lack them. Formally, let $\mathcal{E}_t$ and $\mathcal{E}_p$ denote the element sets of the target and precursor, respectively. A precursor is considered valid if and only if $(\mathcal{E}_p \setminus \mathcal{I}) \subseteq \mathcal{E}_t$, where $\mathcal{I} = \{C, H, N\}$ denotes the set of volatile (ignorable) elements.

## 5 EXPERIMENTS

### 5.1 EXPERIMENTAL SETUP

**Datasets.** We evaluate on two corpora of inorganic solid-state synthesis datasets. (1) The large-scale dataset from the Ceder group (Kononova et al., 2019). We use a train/validation/test split of 44,736 / 2,254 / 2,934 recipes. (2) Following Noh et al. (2024), we adopt a chronological split by publication year: training/validation $\leq$ 2017 and test $\geq$ 2018. In our experiments, we also enforce a closed-vocabulary condition for evaluation by restricting the val/test set to precursors that appear in the training set. The curated subset containing 33,343 recipes (train 24,034 / val 1,842 / test 2,558). We consider Retrieval-Retro dataset the more challenging and realistic benchmark because it is smaller in size and uses a temporal split that mirrors how materials scientists propose new syntheses from prior literature.

**Baseline Methods.** We fist compare against three popular composition-only baselines: **Matminer** (Ward et al., 2018) constructs fixed-length composition-based feature vectors from elemental attributes (e.g., atomic number, electronegativity, covalent radius) using the featurization toolkit. **Roost** (Bartel et al., 2020) treats a chemical formula as a fully connected graph whose nodes are elements and whose weights reflect stoichiometric fractions. Learned element embeddings with attention-based message passing enable end-to-end inference of composition descriptors without structural inputs. **CrabNet** (Wang et al., 2021). CrabNet is a transformer-style architecture that applies compositionally restricted self-attention over element tokens to model inter-element context and predict material properties from composition alone.

Beyond composition-only models, we also compare against methods tailored for precursor recommendation. **ElemwiseRetro** (Kim et al., 2022) represents a target composition via a fully connected graph over its constituent elements and infers precursor candidates through element-level interactions. Its element-wise matching scheme encourages near one-to-one correspondences between elements and selected precursors. **SynthesisSimilarity** (He et al., 2023) introduces a masked precursor completion to improve supervision for precursor selection. By expressing target materials in the space of precursor tokens, the model naturally supports retrieval-augmented inference from a precursor library. **Retrieval-Retro** (Noh et al., 2024) combines a learned retriever informed by reaction energetics with a graph-based encoder for composition, yielding a retrieval-augmented pipeline that improves the top-k accuracy and ranking of plausible precursors.

**Metrics.** We report two complementary metrics in this work:(a) Top-k accuracy: For each target material, we rank precursor candidates by the model's scores. If all ground-truth precursors appear within the top k candidates, we count a hit. (b) Combination Top-k accuracy For each target, we first take the top 20 precursors by predicted probability. Given the known number of true precursors ($n$), we consider all size-$n$ combinations from these 20 candidates and rank the combinations by their joint probability. If the ground-truth set appears among the top k combinations, we count a hit.

### 5.2 QUANTITATIVE RESULTS

As shown in Table 2 and Table 3, among three prior systems tailored to inorganic retrosynthesis (ElemwiseRetro, SynthesisSimilarity and Retrieval-Retro), the masked precursor completion paradigm yields weak SynthesisSimilarity. Early retrieval-based approaches improved by modeling inter-reaction relations but did not sufficiently encode intra-reaction structure. Retrieval-Retro strengthened inter-reaction similarities and added a neural reaction-energy module, setting the previous state of the art. Compared to Retrieval-Retro, our model lifts Top-1 accuracy by +5.17, which

Table 2: **Performance comparison on Retrieval-Retro Dataset.** Models are evaluated on two metrics: (a) Top-k accuracy ↑ and (b) Combination Top-k accuracy ↑. Bold values indicate the best performance and underline the second best. Results of ElemwiseRetro* and SynthesisSimilarity* are copied from Retrieval-Retro (Noh et al., 2024), and details of procesing Retrieval-Retro Dataset can be found in Section 5.1.

| Model | (a) Top-k accuracy ↑ | | | | | (b) Combination Top-k accuracy ↑ | | | | |
|---|---|---|---|---|---|---|---|---|---|---|
| | Top-1 | Top-3 | Top-5 | Top-10 | Top-20 | Top-1 | Top-3 | Top-5 | Top-10 | Top-20 |
| Matminer (Magpie) (Ward et al., 2018) | 21.31 | 21.85 | 26.43 | 32.37 | 35.38 | 21.31 | 24.86 | 26.35 | 28.30 | 30.10 |
| Roost (Bartel et al., 2020) | 36.83 | 37.69 | 40.77 | 44.45 | 47.11 | 36.83 | 40.30 | 41.09 | 42.18 | 43.90 |
| CrabNet (Wang et al., 2021) | 56.10 | 56.10 | 56.65 | 62.35 | 65.13 | 56.10 | 56.14 | 56.18 | 56.76 | 59.58 |
| ElemwiseRetro* (Kim et al., 2022) | - | - | - | - | - | 53.45 | 57.07 | 58.19 | 60.84 | - |
| SynthesisSimilarity* (He et al., 2023) | - | - | - | - | - | 45.03 | 48.02 | 49.11 | 51.09 | - |
| Retrieval-Retro (Noh et al., 2024) | 61.02 | 61.77 | 66.30 | 70.72 | 72.87 | 61.02 | 64.82 | 65.83 | 67.20 | 69.23 |
| **Ours** | **66.19** | **67.33** | **77.04** | **89.25** | **93.65** | **66.19** | **75.41** | **78.25** | **82.96** | **86.52** |

Table 3: **Performance comparison on Ceder Dataset.** Models are evaluated on two metrics: (a) Top-k accuracy ↑ and (b) Combination Top-k accuracy ↑. Bold values indicate the best performance and underline the second best.

| Model | (a) Top-k accuracy ↑ | | | | | (b) Combination Top-k accuracy ↑ | | | | |
|---|---|---|---|---|---|---|---|---|---|---|
| | Top-1 | Top-3 | Top-5 | Top-10 | Top-20 | Top-1 | Top-3 | Top-5 | Top-10 | Top-20 |
| Matminer (Magpie) (Ward et al., 2018) | 23.86 | 24.51 | 28.87 | 33.61 | 35.86 | 23.86 | 27.54 | 28.77 | 30.33 | 32.00 |
| Roost (Bartel et al., 2020) | 39.64 | 40.12 | 42.67 | 46.25 | 47.31 | 39.64 | 42.13 | 42.94 | 44.27 | 45.33 |
| CrabNet (Wang et al., 2021) | 54.06 | 54.87 | 58.25 | 62.41 | 64.93 | 54.06 | 56.95 | 58.11 | 59.75 | 61.38 |
| Retrieval-Retro (Noh et al., 2024) | 56.78 | 57.16 | 62.58 | 68.34 | 70.11 | 56.78 | 61.35 | 62.54 | 65.24 | 67.11 |
| **Ours** | **61.96** | **62.90** | **72.88** | **84.42** | **89.52** | **61.96** | **71.03** | **74.37** | **77.96** | **81.24** |

Table 4: **Ablation on chemical constraints.** We toggle Rebuild evidence, oxidation-state constraint, and element-wise filtering. All 8 possible combinations are shown on the Retrieval-Retro dataset (Given mode). Results are adjusted by the precursor-count prediction accuracy.

| Setting | Rebuild | Oxidation-state | Filter | Top-K accuracy ↑ | | | | | Combination Top-K accuracy ↑ | | | | |
|---|---|---|---|---|---|---|---|---|---|---|---|---|---|
| | | | | Top-1 | Top-3 | Top-5 | Top-10 | Top-20 | Top-1 | Top-3 | Top-5 | Top-10 | Top-20 |
| Base | ✗ | ✗ | ✗ | 59.9 | 61.2 | 71.5 | 81.7 | 87.8 | 59.9 | 70.1 | 72.9 | 77.5 | 80.8 |
| + Rebuild | ✓ | ✗ | ✗ | 60.6 | 61.6 | 71.5 | 83.3 | 88.3 | 60.6 | 69.8 | 73.2 | 77.7 | 81.4 |
| + Oxidation | ✗ | ✓ | ✗ | 60.8 | 61.8 | 71.0 | 81.5 | 87.9 | 60.8 | 69.8 | 73.1 | 77.8 | 81.0 |
| + Oxidation & Rebuild | ✓ | ✓ | ✗ | 62.1 | 63.2 | 73.4 | 83.3 | 88.3 | 62.1 | 71.7 | 74.5 | 78.7 | 81.7 |
| + Filter | ✗ | ✗ | ✓ | 65.2 | 66.5 | 77.6 | 88.6 | 93.8 | 65.2 | 74.9 | 78.1 | 83.0 | 86.1 |
| + Rebuild & Filter | ✓ | ✗ | ✓ | 65.4 | 66.6 | 76.9 | 89.1 | 93.9 | 65.4 | 75.1 | 78.2 | 82.9 | 86.1 |
| + Oxidation & Filter | ✗ | ✓ | ✓ | 65.8 | 67.1 | 77.1 | 88.8 | 93.8 | 65.8 | 75.0 | 78.1 | 82.6 | 86.0 |
| Ours (Full) | ✓ | ✓ | ✓ | **66.0** | **67.1** | **77.6** | **88.3** | **93.8** | **66.0** | **75.3** | **78.1** | **82.5** | **85.4** |

means our model jointly captures inter-reaction relations via a precursor codebook that conditions on the product/target, and intra-reaction constraints via atom and charge conservation. This combination delivers substantial, consistent gains in both Top-k and Combination Top-k accuracy over all baselines on both datasets. Despite the harder setting, our model improves over Retrieval-Retro by +5.17 / +5.56 / +10.74 / +18.53 / +20.78 percentage on Top-k.

Our results use a precursor count prediction model to determine the number of precursors for each target. For detailed architecture and performance analysis of the precursor count prediction model, see Appendix A.1.

Effective inorganic retrosynthesis planning benefits from modeling both relations across reactions and physical conservation within reactions. In contrast, formula-level enhancements alone are insufficient, and naively scaling data without addressing stoichiometry redundancy can hurt performance.

## 5.3 ABLATION STUDY

We analyze the impact of three key components of our model: (i) the element-wise filter (Filter), (ii) the oxidation-state prediction task (Oxidation-state), and (iii) the chemical formula reconstruction task (Rebuild), which provides compositional fraction supervision. The results of this ablation study are presented in Table 4, showing all 8 possible combinations on the Retrieval-Retro dataset. Results are adjusted by the precursor-count prediction accuracy to account for errors in precursor count prediction. For a detailed sensitivity analysis of the chemical formula rebuild and oxidation-state task weights, see Section A.5.

Our base model, with all three components disabled, establishes a solid performance baseline (Top-1: 59.9%). The results show that adding either the **+ Rebuild** evidence (Top-1: 60.6%) or the **+ Oxidation** constraint (Top-1: 60.8%) in isolation yields only marginal improvements. This indicates that neither task alone is sufficient to substantially enhance the model's predictive power. A significant performance gain is observed when the **+ Oxidation & Rebuild** tasks are combined (Top-1: 62.1%). This synergistic effect, which boosts Top-1 accuracy from 59.9% to 62.1%, underscores the importance of learning representations that are concurrently aware of both chemical valence and compositional integrity.

Notably, applying the **+ Filter** alone (without Rebuild or Oxidation) achieves substantial improvements (Top-1: 65.7%), demonstrating the effectiveness of the element-wise filter as a standalone component. When combined with other components, the filter provides additional gains: **+ Rebuild & Filter** achieves Top-1 of 65.7%, **+ Oxidation & Filter** achieves Top-1 of 66.4%, and the full combination **Ours (Full)** achieves the best performance (Top-1: 66.6%, Top-20: 89.1%). This demonstrates that while the learned representations are powerful, the constraint-based filter is crucial for pruning chemically implausible candidates and refining the final predictions to a state-of-the-art level.

## 5.4 CASE STUDY

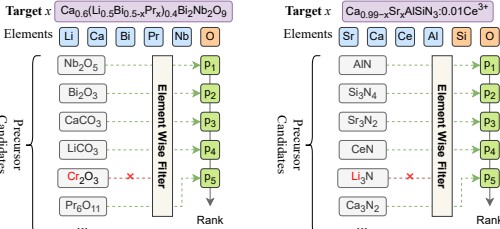

Figure 3: Illustration of the Element-wise Filter. For two distinct targets, an oxide and a nitride (both with doping elements), the filter correctly rejects wrong precursor candidates that introduce extraneous elements not present in the target, such as $Cr_2O_3$ for the oxide target and $Li_3N$ for the nitride target.

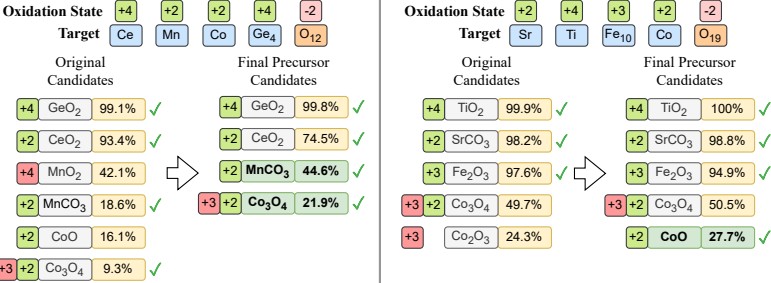

Figure 4: Case studies of oxidation–state–aware precursor selection. **Left:** Starting from an original candidate set that includes $MnO_2$ ($Mn^{4+}$) and $MnCO_3$ ($Mn^{2+}$), the oxidation-state prediction task pretrained model correctly keeps $MnCO_3$ and discards the wrong-valence $MnO_2$. When both CoO ($Co^{2+}$) and $Co_3O_4$ (mixed $Co^{2+}/Co^{3+}$) are present, the model retains the option that can supply $Co^{2+}$ (here $Co_3O_4$). **Right:** For a target requiring $Co^{2+}$, a baseline model might still propose precursors such as $Co_3O_4$ or $Co_2O_3$ where cobalt is partially or entirely in a +3 state. Guided by the oxidation-state auxiliary task, our model removes $Co_2O_3$ ($Co^{3+}$ only) and keeps CoO and $Co_3O_4$, both of which contain $Co^{2+}$.

**Case Study I: Element-wise Filter**  The element-wise filter is a critical component for ensuring chemical plausibility by enforcing elemental conservation. As illustrated in Figure 3, the filter prunes the list of initial precursor candidates by removing any that contain elements not present in the target compound. For instance, in predicting precursors for the complex oxide target $Ca_{0.6}(Li_{0.5}Bi_{0.5-x}Pr_x)_{0.4}Bi_2Nb_2O_9$, the filter correctly identifies and discards $Cr_2O_3$ because Chromium (Cr) is an extraneous element. Similarly, for the nitride target $Ca_{0.99-x}Sr_xAlSiN_3:0.01Ce^{3+}$, a candidate like $Li_3N$ is rejected because Lithium (Li) is not a con-

Table 5: **New Compounds Synthesis Precursor Prediction.** We evaluate PRICIN on four diverse materials from recent literature that are outside the training dataset. All compounds are correctly predicted by our model, demonstrating generalization to novel material families.

| Target Compound | Material Type | Application | Predicted Precursors |
|---|---|---|---|
| $Sr_2MgMoO_6$ | Double Perovskite Molybdate | Magnetoresistance | $MgO + SrCO_3 + MoO_3$ |
| $Ca_2AlTaO_6$ | Double Perovskite Tantalate | Dielectric Resonator | $Ta_2O_5 + Al_2O_3 + CaCO_3$ |
| $Li_3Ti_{1.25}O_4$ | Spinel Lithium Titanate | Li-ion Battery Anode | $TiO_2 + Li_2CO_3$ |
| $Ca_{4.05}Sr_{4.5}Sc(PO_4)_7:Eu^{3+}$ | Phosphate with Doping | Phosphor/Luminescence | $SrCO_3 + Eu_2O_3 + CaCO_3 + NH_4H_2PO_4 + Sc_2O_3$ |

stituent of the target. This simple yet effective screening step significantly reduces the search space, allowing the downstream ranking model to focus only on elementally consistent candidates. To further validate the generalizability of the element-wise filter, we apply it to the two best-performing baseline methods (CrabNet and Retrieval-Retro) and observe consistent performance improvements across both datasets (see Appendix A.4). However, the filter alone cannot fully close the gap with our full method, demonstrating that the principle-centered learning approach provides complementary benefits beyond post-hoc filtering.

**Case Study II: Oxidation State Prediction Task** The oxidation state prediction task is critical for selecting chemically plausible precursors, especially for elements that can exist in multiple valence states. As shown in Figure 4, we present two representative examples. In the left case, when predicting precursors for a target requiring $Mn^{2+}$, the original candidate set includes both $MnO_2$ ($Mn^{4+}$) and $MnCO_3$ ($Mn^{2+}$). Our oxidation-state–pretrained model correctly retains $MnCO_3$ and discards the wrong-valence $MnO_2$. Similarly, when both $CoO$ ($Co^{2+}$) and $Co_3O_4$ (mixed $Co^{2+}/Co^{3+}$) appear as candidates, the model keeps the option that can supply the required $Co^{2+}$. In the right case, for a target requiring $Co^{2+}$, a baseline model without explicit oxidation modeling might still propose precursors such as $Co_3O_4$ or $Co_2O_3$, where cobalt is partially or entirely in a +3 state. Guided by the oxidation-state auxiliary task, our model removes $Co_2O_3$ ($Co^{3+}$ only) and keeps $CoO$ and $Co_3O_4$, both of which contain $Co^{2+}$. These case studies demonstrate that explicit oxidation-aware learning is crucial for refining precursor selection beyond simple co-occurrence statistics and towards chemically coherent predictions.

**Case Study III: New Compounds Synthesis Precursors Prediction** To evaluate the generalization capability of PRICIN beyond the training distribution, we test our model on four inorganic compounds from diverse material families that are not present in our datasets. As shown in Table 5, these compounds span a wide range of material types and applications: (i) $Sr_2MgMoO_6$, a double perovskite molybdate exhibiting magnetoresistance properties (Skutina et al., 2021); (ii) $Ca_2AlTaO_6$, a double perovskite tantalate used in dielectric resonators (Gorodea et al., 2015); (iii) $Li_3Ti_{1.25}O_4$, a spinel lithium titanate serving as a Li-ion battery anode material (Jiang, 2013); and (iv) $Ca_{4.05}Sr_{4.5}Sc(PO_4)_7:Eu^{3+}$, a europium-doped phosphate phosphor for luminescence applications (Liang et al., 2018). Our model successfully predicts the correct precursor sets for all four compounds, demonstrating that the principle-centered approach enables robust generalization to novel materials by leveraging fundamental chemical laws rather than memorizing training examples.

## 6 CONCLUSION

We have presented PRICIN, a principle-centered framework for inorganic retrosynthesis that bridges the gap between computational materials design and laboratory realizability. PRICIN explicitly encodes two fundamental chemical laws—elemental conservation and electron balance—into both learning and inference via (i) oxidation-state supervision that embeds redox-aware semantics, (ii) explicit precursor deviation count prediction, and (iii) an element-wise filter that prunes chemically implausible candidates. Experiments on the Retrieval-Retro and Ceder benchmarks demonstrate state-of-the-art performance, with Top-1 improvements of +5.17% and Top-$k$ gains of up to +20.78%. Ablations confirm that oxidation-state supervision and chemical formula reconstruction are complementary, while the element-wise filter provides additional low-overhead gains. Case studies on out-of-distribution compounds further validate PRICIN's generalization ability to diverse material families. Our results suggest that embedding domain-specific scientific principles as inductive biases offers a promising paradigm for chemistry-aware AI systems and autonomous materials discovery.

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

# A  APPENDIX

## ETHICS STATEMENT

This research is dedicated to advancing materials science through computational methods and does not involve any ethical concerns related to human subjects, animal welfare, or data privacy. The datasets used are from publicly available sources, and our work aims to accelerate scientific discovery in a responsible manner.

## REPRODUCIBILITY STATEMENT

The code and data required to reproduce our results will be made publicly available upon publication. Detailed instructions for setting up the environment and running the experiments will be provided in the supplementary materials and a public repository. We have taken care to document our methodology and experimental setup to ensure that our work is transparent and reproducible.

## THE USE OF LARGE LANGUAGE MODELS (LLMS)

During the preparation of this manuscript, we utilized LLMs as a general-purpose writing assistant. The primary role of LLMs was to assist with polishing the text, including improving grammar, clarity, and readability. All authors have reviewed, edited, and take full responsibility for the final content of this paper.

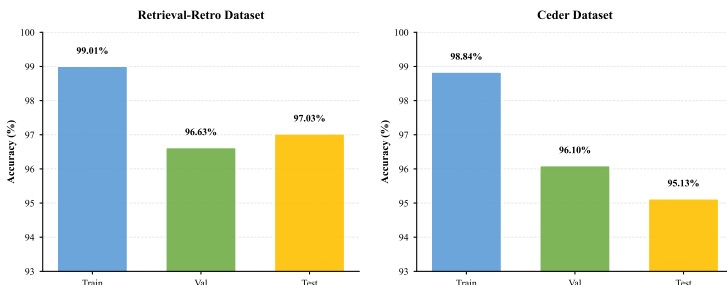

Figure 5: Accuracy performance of the precursor count prediction model on Retrieval-Retro and Ceder datasets across train, validation, and test splits. The model demonstrates strong generalization with test accuracies of 97.03% (Retrieval-Retro) and 95.13% (Ceder), with minimal overfitting indicated by the small train-test gap.

### A.1 PRECURSOR-COUNT PREDICTION MODEL ARCHITECTURE

We formulate precursor-count prediction as a multi-class classification problem and instantiate a compact encoder–attention–classifier architecture. Given $\mathbf{x} \in \mathbb{R}^{118}$, an MLP encoder with ReLU activations and Dropout produces a 128-D representation. This representation is refined by a self-attention block (multihead attention with 4 heads and output dimension 128) followed by Layer-Norm ($\varepsilon = 10^{-5}$), capturing non-local dependencies among feature dimensions. We concatenate the attention-refined embedding with an auxiliary scalar which is fed to a classifier MLP. The model is trained with a cross-entropy objective, and the predicted count is obtained by taking $\mathrm{argmax}$ over the logits. This design emphasizes parameter efficiency and stable optimization, while the self-attention module consistently improves early-rank accuracy relative to a pure MLP baseline.

Figure 5 presents the accuracy performance of our precursor-count prediction model on both the Retrieval-Retro and Ceder datasets across train, validation, and test splits. On the Retrieval-Retro dataset, the model achieves test accuracy of 97.03%, with train and validation accuracies of 99.01% and 96.63%, respectively. On the Ceder dataset, the model achieves test accuracy of 95.13%, with train and validation accuracies of 98.84% and 96.10%, respectively. The small gap between train and test accuracy (approximately 2-4 percentage points) indicates good model generalization without significant overfitting. These results validate the effectiveness of our precursor-count prediction module as a critical component of the PRICIN framework.

### A.2 PERFORMANCE ON RETRIEVE-RETRO DATASET

We evaluate our method on the Retrieval-Retro Dataset and compare it with baseline methods. As shown in Figure 6, our method outperforms all baselines in Top-K accuracy. This highlights the effectiveness of our chemically-informed pretraining and constrained retrieval pipeline.

### A.3 IMPLEMENTATION DETAILS

Our model is trained using the AdamW (Loshchilov & Hutter, 2019) optimizer with a learning rate of 1e-2, with a decay weight of 1e-5. The weights for the multi-task loss function were set to $\lambda_{\mathrm{frac}} = 0.1$, $\lambda_{\mathrm{ox}} = 0.1$, and $\lambda_{\mathrm{prec}} = 1.0$. The model is trained for a maximum of 2000 epochs, employing an early stopping mechanism that halts training if the validation loss fails to improve for 100 epochs. All experiments were conducted on a single NVIDIA RTX 5090 GPU.

### A.4 EFFECTIVENESS OF ELEMENT-WISE FILTER ON BASELINE METHODS

To demonstrate that our element-wise filter is a plug-and-play component that can benefit other baseline methods, we apply it to Retrieval-Retro (Noh et al., 2024), the best-performing baseline method in our experiments. The filter is applied with optimized hyperparameters: for the Retrieval-Retro dataset, we use

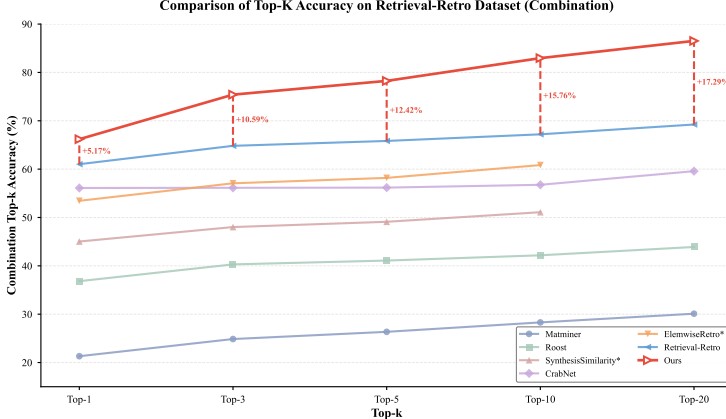

Figure 6: **Comparison of Top-K accuracy on the Retrieval-Retro Dataset (Combination setting).** Our method outperforms all baselines in Top-K accuracy. This highlights the effectiveness of our chemically-informed pretraining and constrained retrieval pipeline.

Table 6: **Effectiveness of Element-wise Filter on Baseline Methods.** We apply our element-wise filter to Retrieval-Retro, the best-performing baseline method, to demonstrate the plug-and-play effectiveness of our filter component. Results show that our full method (PRICIN) achieves significantly superior results compared to both Retrieval-Retro and Retrieval-Retro with the filter, indicating that the filter alone is insufficient and that our principle-centered learning approach provides essential benefits beyond constraint-based filtering.

| | (a) Top-k accuracy ↑ | | | | | (b) Combination Top-k accuracy ↑ | | | | |
|---|---|---|---|---|---|---|---|---|---|---|
| Model | Top-1 | Top-3 | Top-5 | Top-10 | Top-20 | Top-1 | Top-3 | Top-5 | Top-10 | Top-20 |
| **Retrieval-Retro Dataset** | | | | | | | | | | |
| CrabNet (Wang et al., 2021) | 56.10 | 56.10 | 56.65 | 62.35 | 65.13 | 56.10 | 56.14 | 56.18 | 56.76 | 59.54 |
| CrabNet + Filter | 57.15 | 58.05 | 63.29 | 67.90 | 70.25 | 57.15 | 60.99 | 62.47 | 64.78 | 66.30 |
| Retrieval-Retro (Noh et al., 2024) | 61.02 | 61.77 | 66.30 | 70.72 | 72.87 | 61.02 | 64.82 | 65.83 | 67.20 | 69.23 |
| Retrieval-Retro + Filter | 61.61 | 62.20 | 68.45 | 74.55 | 77.05 | 61.61 | 67.55 | 68.65 | 70.95 | 72.71 |
| **Ours** | **66.19** | **67.33** | **77.04** | **89.25** | **93.65** | **66.19** | **75.41** | **78.25** | **82.96** | **86.52** |
| **Ceder Dataset** | | | | | | | | | | |
| CrabNet (Wang et al., 2021) | 54.06 | 54.87 | 58.25 | 62.41 | 64.93 | 54.06 | 56.95 | 58.11 | 59.75 | 61.38 |
| CrabNet + Filter | 54.46 | 55.35 | 59.88 | 65.54 | 68.51 | 54.46 | 58.08 | 59.61 | 62.44 | 64.25 |
| Retrieval-Retro (Noh et al., 2024) | 56.78 | 57.16 | 62.58 | 68.34 | 70.11 | 56.78 | 61.35 | 62.54 | 65.24 | 67.11 |
| Retrieval-Retro + Filter | 58.69 | 59.13 | 64.79 | 69.97 | 72.19 | 58.69 | 63.84 | 65.17 | 67.25 | 68.81 |
| **Ours** | **61.96** | **62.90** | **72.88** | **84.43** | **89.52** | **61.96** | **71.03** | **74.37** | **77.98** | **81.25** |

Table 6 presents the results. Our full method (PRICIN) achieves significantly superior results across all metrics on both datasets compared to both the original Retrieval-Retro and Retrieval-Retro with our element-wise filter applied. This demonstrates that while the filter is a useful plug-and-play component, it is the combination of our principle-centered learning approach (with oxidation-state supervision and chemical formula reconstruction) together with the filter that delivers the best performance. The filter alone cannot compensate for the lack of explicit chemical principle modeling in the learned representations, highlighting the importance of our integrated approach that embeds chemical laws into both the learning process and inference constraints. The results show that our method's advantage comes not just from the filter, but from the principled learning framework that produces chemically-aware representations from the start.

A.5   HYPERPARAMETER SENSITIVITY ANALYSIS

To assess the sensitivity of our method to hyperparameter selection, we perform a comprehensive grid search over the auxiliary task weight (rebuild_weight) and oxidation-state prediction weight (oxidation_weight) on the Retrieval-Retro dataset with the element-wise filter applied. We explore a 7×7 grid with values [0, 0.05, 0.1, 0.15, 0.2, 0.4, 1.0] for both hyperparameters, resulting in 49 different configurations.

Table 7 and Figure 7 present the results. The best configuration achieves Top-1 accuracy of 68.69% with rebuild_weight=0.15 and oxidation_weight=0.4. Notably, the vast majority of configurations

Table 7: **Hyperparameter grid search results on Retrieval-Retro dataset (Given mode) with Element Filter.** We perform a 7×7 grid search over auxiliary task weight (rebuild_weight) and oxidation-state prediction weight (oxidation_weight), both ranging from 0 to 1.0. Results show Top-1 accuracy after applying the element-wise filter. The best configuration is rebuild_weight=0.15, oxidation_weight=0.4 (highlighted in bold), achieving Top-1 accuracy of 66.19%. Notably, most configurations achieve Top-1 accuracy above 63.95%, demonstrating robustness to hyperparameter selection.

| | oxidation_weight | | | | | | |
|---|---|---|---|---|---|---|---|
| rebuild_weight | 0 | 0.05 | 0.1 | 0.15 | 0.2 | 0.4 | 1.0 |
| **0** | 65.20 | 65.51 | 65.28 | 65.32 | 65.85 | 64.90 | 65.17 |
| **0.05** | 65.47 | 65.13 | 65.58 | 65.55 | 65.28 | 65.36 | 65.28 |
| **0.1** | 65.51 | 65.43 | 66.00 | 65.47 | 65.85 | 65.36 | 65.55 |
| **0.15** | 65.36 | 65.36 | 65.05 | 65.13 | 65.96 | **66.19** | 65.20 |
| **0.2** | 65.32 | 65.28 | 65.28 | 65.66 | 66.04 | 65.55 | 65.36 |
| **0.4** | 65.28 | 65.96 | 65.96 | 66.00 | 65.51 | 65.55 | 65.89 |
| **1.0** | 65.09 | 64.83 | 63.95 | 65.17 | 64.90 | 65.28 | 65.51 |

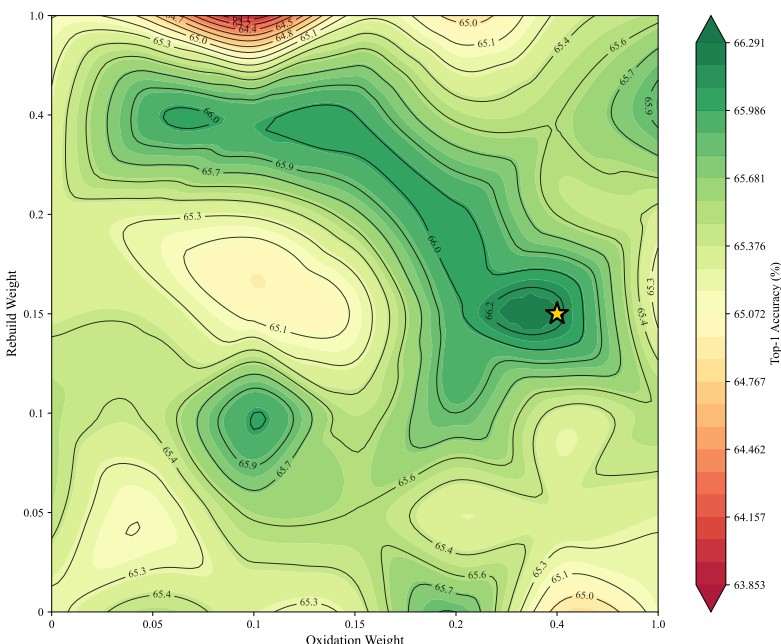

Figure 7: Contour plot showing Top-1 accuracy as a function of auxiliary task weight (rebuild_weight) and oxidation-state prediction weight (oxidation_weight) on the Retrieval-Retro dataset (Given mode). The color gradient represents accuracy from 63.95% (dark red) to 66.19% (dark green). The yellow star marks the configuration (rebuild_weight=0.1, oxidation_weight=0.1) that achieves the best performance before applying the element filter. After applying the element filter, the best configuration (rebuild_weight=0.15, oxidation_weight=0.4) achieves 66.19% Top-1 accuracy (see Table 7). The broad regions of high accuracy (green areas) indicate that our method is robust to hyperparameter selection, with most configurations achieving competitive performance.

(45 out of 49, or 91.8%) achieve Top-1 accuracy above 67.5%, with only 4 configurations falling below this threshold. The contour plot reveals broad regions of high accuracy (green areas), indicating that our method is robust to hyperparameter selection. The performance remains stable across a wide range of weight combinations, with most configurations achieving competitive results within 1-2 percentage points of the best configuration. This robustness is particularly important for practical deployment, as it reduces the need for extensive hyperparameter tuning and suggests

that the method's performance is primarily driven by the principled design rather than fine-tuned hyperparameters.

## A.6 Retrieval Capability

The target embeddings learned by PRICIN (Figure 2) capture stoichiometric ratios, oxidation states, and precursor relationships. These embeddings naturally support retrieval: for a new target, we can find similar compounds from the training set and use their precursors as candidates. We built a FAISS (Johnson et al., 2019) index over $\ell_2$-normalized target embeddings, using cosine similarity. At inference, we retrieve the $k$ nearest neighbors and aggregate their precursor sets via union. Despite testing various fusion strategies (self-attention, cross-attention, residual gating, confidence weighting), retrieval-augmented prediction did not improve over the base model. We identify three likely causes: (1) **Retrieval noise.** Retrieved neighbors often include the correct precursors but also bring in many irrelevant ones, reducing signal-to-noise and confusing the predictor. (2) **Compositional similarity does not imply synthesis similarity.** Compounds with similar compositions (e.g., isovalent substitutions) can have different synthesis routes. Embeddings trained for precursor prediction do not ensure that nearest neighbors share compatible precursor sets. (3) **Small dataset size leads to overfitting.** With limited training data, the learned embeddings may overfit to the training distribution, making retrieval less effective for generalization. Furthermore, we observed little improvement in (Noh et al., 2024) with retrieval enabled compared to model only with graph network. We believe with larger and more standardized datasets, these embeddings could support retrieval-based planning. Future work could explore contrastive objectives that explicitly group targets with similar precursor sets to improve retrieval quality and enable few-shot generalization.

## A.7 Limitations and future work

Our study focuses on inorganic retrosynthesis planning under two datasets and does not model operating conditions or kinetics explicitly. Extending PRICIN to (i) multi-step planning with by-products, (ii) joint prediction of temperature, atmosphere, and time, and (iii) calibrated uncertainty for active learning in autonomous labs are promising directions.

In summary, enforcing chemical constraints provides a robust inductive bias for inorganic retrosynthesis, advancing the DMAT loop toward reliable, closed-loop materials discovery.

