# OpenReview forum: "PRICIN: Principle-Centered Inorganic Retrosynthesis"
_ICLR.cc/2026/Conference — ICLR 2026 Conference Withdrawn Submission_

### Official Review · Reviewer_Bszy · 2025-10-27

**Soundness:** 3
**Presentation:** 2
**Contribution:** 3
**Rating:** 4
**Confidence:** 2

**Summary:**

This work presents a novel *inorganic retrosynthesis* framework based on a principle-centered approach.

**Strengths:**

- The paper is well-motivated.
- The principle-centered approach is innovative and has strong potential for guiding inorganic retrosynthesis.

**Weaknesses:**

### **Major Comments**
1. **Order Invariance**
   It is unclear whether the proposed method is order-invariant when dealing with different arrangements of elements.

2. **Variable Valence Materials**
   Can the method handle materials with variable valence states? For example, in Fe₃O₄ there are two Fe³⁺ and one Fe²⁺ ions. How does the model represent and process such mixed-valence compounds?

3. **Missing Details on Element-wise Filter**
   Section 4 lacks specific implementation details regarding the “Element-wise Filter.” How are these filters defined and applied? A more detailed description or pseudo-code would help readers understand this component.

4. **Clarity of Figure 2**
   Figure 2 is difficult to interpret. What does the number *118* represent in the figure? Also, in the Task 2 subfigure, the oxidation ground truth also shows *118*. Please clarify its meaning and ensure all symbols are clearly explained.

5. **Number of Predicted Precursors**
   Do all predicted precursor sets have the same number of materials that predicted by the deviation predictor?

6. **Codebook Construction**
   Please elaborate on how the *Oxidation Number Codebook* and *Precursor Codebook* are constructed. Are they predefined, learned jointly with the model, or created from external data?


### **Minor Comments**
- Several citation style issues were found. Please use `\citet{}` instead of `\cite{}`  to maintain consistency with the required citation style.

**Questions:**

see weaknesses

---

> ### Author Response · Authors · 2025-12-01
>
> Dear Reviewer Bszy,
>
> We sincerely thank you for your detailed and constructive feedback. Your comments helped us clarify several important aspects of PRICIN, and we have revised the manuscript accordingly. Below we respond to each point and indicate the corresponding changes in the paper.
>
> ---
>
> > **(W1) Question: Order invariance of the composition representation is not clearly demonstrated**
>
> PRICIN is order-invariant with respect to the arrangement of elements in the chemical formula. Our encoder does **not** process a sequence of element tokens; instead, it takes as input a fixed-length **composition vector** of dimension $N_E = 118$, where each coordinate corresponds to one chemical element in the periodic table and stores its (normalized) fraction in the compound. This representation depends only on the multiset of elements and their stoichiometric fractions, not on the textual order in which the formula is written. We note that if one were to adopt an alternative encoder that directly consumes ordered element tokens or sequence-like representations, then the input order could become relevant; in PRICIN, however, the composition featurizer is deliberately designed to be order invariant.
>
> > **(W2) Question: Support for variable-valence materials such as $\mathrm{Fe_3O_4}$ is not sufficiently clarified**
>
> PRICIN is explicitly designed to handle **variable valence (mixed-oxidation-state) materials**. In **Section 3 (Electron balance)** and **Section 4.2 (Task 2: Oxidation State Prediction)**, we model oxidation states at the level of element–oxidation-state pairs ("species"). For each element $E$ and oxidation state $(z \in \{-7, \dots, +7\})$, we allocate one entry in the **Oxidation State Codebook**$ (C_{\text{ox}})$. Thus, each element has 15 possible valence “slots” in the model.
>
> For $\mathrm{Fe_3O_4}$ in particular, we treat it as $\mathrm{FeO + Fe_2O_3}$, so that the Fe atoms appear in both $(+2)$ and $(+3)$ oxidation states. In our construction of the ground-truth oxidation distribution, 2/15 oxidation states for Fe are assigned, so that the corresponding representation naturally fuses information from both $Fe^{2+}$ and $Fe^{3+}$. The model then predicts a probability distribution over all 15 possible oxidation states of Fe, and is trained so that the predicted distribution matches this mixed-valence target.
>
> > **(W3) Question: Implementation details of the Element-Wise Filter are insufficiently described**
>
> To better explain this component, we have introduced a dedicated subsection **4.4 Element-Wise Filter**, where we describe the motivation and the exact formulation. The key paragraph added is:
>
> > Existing methods implicitly learn precursor selection over the entire candidate set. Although this approach achieves reasonable prediction performance, the top-(k) ranked candidates may still contain **chemically invalid choices** that violate elemental conservation principles. To address this limitation, we introduce an **Element-Wise Filter** as a post-processing step during inference. As established in our discussion of elemental conservation in Section 3, a valid precursor must satisfy the elemental conservation constraint, i.e., its non-volatile elements should form a subset of the target’s elements. However, solid-state synthesis typically proceeds at elevated temperatures (often exceeding 800 °C), under which certain elements—specifically carbon $C$, hydrogen $H$, and nitrogen $N$—undergo thermal decomposition and are released as gaseous byproducts $CO_2$, $H_{2}O$, and $NO_x$/$NH_3$. Consequently, precursors containing these elements can legitimately contribute to targets that lack them. Formally, let $\mathcal{E}_t$ and $\mathcal{E}_p$ denote the element sets of the target and precursor, respectively. A precursor is considered valid if and only if $(\mathcal{E}_p \setminus \mathcal{I}) \subseteq \mathcal{E}_t$, where $\mathcal{I} = \{\text{C}, \text{H}, \text{N}\}$ denotes the set of volatile (ignorable) elements.
>
> Furthermore, to more thoroughly investigate the impact of the Element-Wise Filter, we also evaluate it on the two baseline methods, and report the results in **Appendix A.4 (Effectiveness of Element-Wise Filter on Baseline Methods)**.
>
> > **(W4) Question: The meaning of “118” and related notation in Figure 2 is unclear and notational consistency is lacking**
>
> We thank you for pointing out this ambiguity. The number **118** in the original figure corresponds to $N_E$ = 118, the total number of chemical elements considered in our composition vector. To make this clearer, we have updated **Figure 2**. Furthermore, we **unify the letters and symbols** used in the methods text and in **Figure 2**, so that notation is consistent across equations and figures and easier to follow. We believe these changes significantly improve the clarity and readability of the figure.

---

> ### Author Response · Authors · 2025-12-01
>
> > **(W5) Question: Number of predicted precursors**
>
> Our pipeline uses the **deviation classifier** to predict the number of precursorsn for each target. The classifier estimates the deviation $\Delta = m - E_{\text{non}}$ from the number of non-metal elements, with $\Delta \in \{-2, -1, 0, +1, +2\}$. We then select the top-$\hat{m}$ precursors after applying the Element-Wise Filter.
>
> Thus, for a given target, the predicted precursor set contains $\hat{m}$ materials as determined by the count predictor; however, $\hat{m}$ may differ from the **ground-truth** count $m$ when the classifier is wrong. On our two test sets, the precursor-count model achieves about **97%** accuracy on Retrieval-Retro and **95%** on the Ceder set, so only roughly **2–5%** of targets have a mismatched count.
>
> > **(W6) Question: Construction of the Oxidation Number Codebook and Precursor Codebook is not clearly explained**
>
> Both the **Oxidation State Codebook** $C_{\text{ox}}$ and the **Precursor Codebook** $C_{\text{prec}}$ are implemented as **learnable embedding matrices**, analogous to `nn.Embedding` layers:
>
> * $C_{\text{ox}}$ contains one embedding for each (element, oxidation) pair, with 15 possible oxidation states (−7 to +7) for each of the $N_E$ elements.
> * $C_{\text{prec}}$ contains one embedding vector for each precursor in the library of $N_p$ precursors observed in the training data.
>
> In both cases, the embeddings are **randomly initialized** and then **learned jointly** with the rest of the model parameters using the combined loss over all tasks.
>
> > **(Q1) Question: Citation style usage**
>
> Thank you for pointing this out. We have carefully proofread the manuscript and standardized the citation style.
>
> ---
>
> We hope that these revisions and clarifications address your concerns and improve the readability and transparency of the manuscript. Thank you again for your insightful comments.
>
> **Sincerely,**
>
> The Authors

---

### Official Review · Reviewer_aKeu · 2025-10-27

**Soundness:** 2
**Presentation:** 2
**Contribution:** 2
**Rating:** 2
**Confidence:** 4

**Summary:**

A principle-centered approach that reformulates precursor planning around two laws: elemental conservation and electron balance.
The paper argues that existing studies focus on precedent-based learning and have weak modeling of chemical principles.

**Strengths:**

- Paper is well written so that I can easily follow the work
- Inorganic retrosynthesis is important problem in the field of materials science

**Weaknesses:**

Lack of Novelty: It appears the only newly added content is the electron-based oxidation number prediction. This is my main concern in this paper that the paper utilizes exisiting ML techniques for inorganic retrosynthesis, and even not new problem formulation

Ablation Study Concerns:
- The filter seems to be the key factor in improving performance in the ablation study. However, there is very little explanation provided about this filter.
- It would be beneficial to apply the filter to all ablated models for a more comprehensive comparison.

Performance of Oxidation vs. Rebuild: There is little performance difference between the oxidation and rebuild components. Can this be explained? The model's main motivation is centered on oxidation, yet it shows only a marginal performance difference from the rebuild component.

**Questions:**

See Weakness section

---

> ### Author Response · Authors · 2025-12-01
>
> Dear Reviewer aKeu,
>
> We sincerely thank you for your careful reading and constructive suggestions. Below we summarize and respond to each of your concerns, and describe the corresponding revisions in the manuscript.
>
> ---
>
> > **(W1) Question: Lack of novelty**
>
> We appreciate this concern and would like to clarify that PRICIN is intended as a **principle-centered framework**, rather than simply "adding an oxidation head" to an existing model.
>
> 1. **Reformulating retrosynthesis around physical laws.**
> We explicitly formulate inorganic retrosynthesis around **elemental conservation** and **electron balance**, and use these as the organizing principles for both representation learning and inference. Instead of purely learning statistical patterns from historical recipes, we structure the model so that its internal representations and post-processing steps must respect these two laws (Section 3 and the three-task design in Figure 2).
>
> 2. **A coordinated four-component design, not just oxidation prediction.**
> Concretely, PRICIN couples four components that are jointly trained and used:
>
> - Chemical Formula Rebuild (Task 1): An auxiliary decoder that reconstructs elemental fractions from the latent embedding, forcing it to encode stoichiometry.
> - Oxidation State Prediction (Task 2): Explicit supervision over element–oxidation pairs via a learned oxidation-state codebook and a dedicated loss, injecting electron-balance awareness.
> - Precursor Number Prediction: A count predictor that classifies the deviation from the number of non-metal elements, rather than relying on ad-hoc thresholding.
> - Element-wise Filter: An inference-time constraint that removes precursors violating elemental conservation (up to volatile C/H/N), thereby aligning final predictions with the target’s element set.
>
> Together, these components are designed to mitigate dataset biases (e.g., overused but chemically inconsistent precursors) and to tie the model’s behavior more closely to chemical principles.
>
> 3. **Explicit oxidation-state supervision as a core modeling primitive.**
> To the best of our knowledge, PRICIN is the first framework to make **oxidation-state supervision a first-class component** of an inorganic retrosynthesis model, via a dedicated oxidation-state codebook and loss (Eq. (4)), rather than relying on composition-only features to implicitly encode redox information. This is highlighted in our contributions and is central to how PRICIN learns its latent space.
>
> 4. **Principle-centered design leads to consistent SOTA gains.**
> The combination of these principle-driven components yields substantial gains on both Retrieval-Retro and Ceder benchmarks. On Retrieval-Retro, PRICIN improves Top-1 accuracy by **+5.17** and Top-20 by **+20.78** over the previous best; on the Ceder benchmark, we see **+5.18** Top-1 and up to **+14.13** Top-20 improvements. These gains are not attributable to a generic architecture change alone; they arise from embedding elemental/electron constraints into both representation learning and the inference pipeline.
>
> > **(W2a) Question: The element-wise filter appears to be a key driver of improvement in the ablation, but its design and operation are not well explained.**
>
> We agree that the filter is an important component and have clarified its mechanism more explicitly in the revised version. To better explain this component, we have introduced a dedicated subsection **4.4 Element-Wise Filter**, where we describe the motivation and the exact formulation. The key paragraph added is:
>
> > Existing methods implicitly learn precursor selection over the entire candidate set. Although this approach achieves reasonable prediction performance, the top-k ranked candidates may still contain **chemically invalid choices** that violate elemental conservation principles. To address this limitation, we introduce an **Element-Wise Filter** as a post-processing step during inference. As established in our discussion of elemental conservation in Section 3, a valid precursor must satisfy the elemental conservation constraint, i.e., its non-volatile elements should form a subset of the target’s elements. However, solid-state synthesis typically proceeds at elevated temperatures (often exceeding 800 °C), under which certain elements—specifically carbon $C$, hydrogen $H$, and nitrogen $N$—undergo thermal decomposition and are released as gaseous byproducts $CO_2$, $H_{2}O$, and $NO_x$/$NH_3$. Consequently, precursors containing these elements can legitimately contribute to targets that lack them. Formally, let $\mathcal{E}_t$ and $\mathcal{E}_p$ denote the element sets of the target and precursor, respectively. A precursor is considered valid if and only if $(\mathcal{E}_p \setminus \mathcal{I}) \subseteq \mathcal{E}_t$, where $\mathcal{I} = \{\text{C}, \text{H}, \text{N}\}$ denotes the set of volatile (ignorable) elements.

---

> ### Author Response · Authors · 2025-12-01
>
> > **(W2b) Question: For a fair ablation, the element-wise filter should be applied to all ablated variants, not only to the final model.**
>
> We agree that a comprehensive comparison is important. In the revised manuscript, **Table 4** explicitly reports all **eight** combinations of Rebuild, Oxidation, and Filter on the Retrieval-Retro setting:
>
> | Setting               |    Top-1 |    Top-3 |    Top-5 |   Top-10 |   Top-20 | Comb Top-1 | Comb Top-3 | Comb Top-5 | Comb Top-10 | Comb Top-20 |
> | --------------------- | -------: | -------: | -------: | -------: | -------: | ---------: | ---------: | ---------: | ----------: | ----------: |
> | Base                  |     59.9 |     61.2 |     71.5 |     81.7 |     87.8 |       59.9 |       70.1 |       72.9 |        77.5 |        80.8 |
> | + Rebuild             |     60.6 |     61.6 |     71.5 |     83.3 |     88.3 |       60.6 |       69.8 |       73.2 |        77.7 |        81.4 |
> | + Oxidation           |     60.8 |     61.8 |     71.0 |     81.5 |     87.9 |       60.8 |       69.8 |       73.1 |        77.8 |        81.0 |
> | + Oxidation & Rebuild |     62.1 |     63.2 |     73.4 |     83.3 |     88.3 |       62.1 |       71.7 |       74.5 |        78.7 |        81.7 |
> | + Filter              |     65.2 |     66.5 |     77.6 |     88.6 |     93.8 |       65.2 |       74.9 |       78.1 |        83.0 |        86.1 |
> | + Rebuild & Filter    |     65.4 |     66.6 |     76.9 |     89.1 |     93.9 |       65.4 |       75.1 |       78.2 |        82.9 |        86.1 |
> | + Oxidation & Filter  |     65.8 |     67.1 |     77.1 |     88.8 |     93.8 |       65.8 |       75.0 |       78.1 |        82.6 |        86.0 |
> | **Ours (Full)**       | **66.0** | **67.1** | **77.6** | **88.3** | **93.8** |   **66.0** |   **75.3** |   **78.1** |    **82.5** |    **85.4** |
>
> Moreover, **Appendix A.4 (Effectiveness of Element-Wise Filter on Baseline Methods)** applies the same element-wise filter to two strong baseline models (CrabNet and Retrieval-Retro). Although these "baseline + filter" variants do improve over the original baselines, **PRICIN still significantly outperforms them**, indicating that the overall performance gains cannot be attributed to the filter alone; the principle-centered training objectives are also crucial.
>
> > **(W3) Question: Oxidation and Rebuild show very similar performance gains. Given that the paper is motivated by oxidation, why does oxidation not clearly outperform Rebuild?**
>
> The main reason is that the two auxiliary tasks guide the model from **different angles**:
>
> * The **chemical formula rebuild** task teaches the model which elements and approximate ratios are reasonable for a given product.
> * The **oxidation-state prediction** task teaches the model which oxidation configurations are reasonable for those elements.
>
> Each task alone only partially narrows down the possible precursor sets; when combined, they jointly constrain both composition and oxidation, which is especially useful under our strict evaluation where the **entire** precursor set must be correct.
>
> To make this clearer, we have aligned the settings (All with precursor number not given) and metrics and now present the ablation results directly. Detailed performance is listed in the reply to (W2b).
>
> ---
>
> We hope these clarifications and revisions address your concerns about novelty and ablation design. Thank you again for your insightful comments, which have helped us improve both the clarity and the rigor of the manuscript.
>
> **Sincerely,**
>
> The Authors

---

### Official Review · Reviewer_QP4Z · 2025-10-30

**Soundness:** 2
**Presentation:** 3
**Contribution:** 2
**Rating:** 2
**Confidence:** 4

**Summary:**

This work focuses on inorganic retrosynthesis, specifically predicting the precursor set for a given target material. The authors incorporate strong chemical rules — particularly oxidation number constraints and element filters based on deviation — to enhance model performance. Extensive experiments are conducted to validate the approach.

**Strengths:**

- The model effectively integrates strong domain knowledge (chemical rules such as elemental conservation and electron balance) directly into the modeling process, leading to improved performance.
- The paper is well-presented and easy to follow.

**Weaknesses:**

- The technical contribution appears limited. Although the proposed framework adopts a multi-task learning setup, Task 1—composition reconstruction—has already been explored in prior work (SynthesisSimilarity, He et al.). The main additions seem to be the chemical rule–based element filtering and oxidation number prediction modules.
 - The paper lacks sufficient explanation of how oxidation number prediction contributes to overall performance and how it is modeled.
- Moreover, using known precursor oxidation states to predict the oxidation states of the target material may restrict generalization to in-distribution targets, since many targets can exhibit multiple or mixed oxidation states. This could lead the model to learn only “easy” oxidation states under overly strong supervision.
-  A sensitivity analysis for each training task would be beneficial to clarify their respective contributions.

**Questions:**

- In the preliminary section, the authors mention by-products. Were by-products explicitly considered during the retrosynthesis task? If so, how were they handled?

- The paper claims strong extrapolation ability to new systems, but there is little explanation or evidence supporting this. There seems to be no specific methodological component designed to improve extrapolation. Since the model mainly builds upon strong chemical-formula-based priors (e.g., oxidation filtering), how does this lead to better extrapolation?

- If oxidation-state prediction is claimed to help handle novel or unseen materials, more detailed discussion is needed: how does this mechanism concretely enable the model to generalize or make accurate predictions for truly new target systems?

---

> ### Author Response · Authors · 2025-12-01
>
> Dear Reviewer QP4Z,
>
> We sincerely thank you for your thoughtful review and constructive feedback. Below we respond to your comments point by point and describe the corresponding clarifications and revisions in the manuscript.
>
> ---
>
> > **(W1) Question: Limited technical contribution; Task 1 is similar to prior work, and the main additions appear to be rule-based element filtering and oxidation-state prediction.**
>
> We appreciate this concern and agree that Task 1 (composition / formula reconstruction) by itself is not novel. Our intention, however, is not to propose a new reconstruction loss, but to build a principle-centered framework in which **elemental conservation** and **electron balance** jointly organize representation learning and inference.
>
> First, the work is not centered on Task 1 alone. Task 1 ("chemical formula rebuild") serves as an auxiliary constraint that forces the target embedding to retain stoichiometric information, but the main contribution lies in how it is coupled with (i) explicit oxidation-state prediction, (ii) a precursor-count prediction head that classifies the deviation from the number of non-metal elements, and (iii) an element-wise filter enforcing conservation at inference. We analyze the empirical relationship between target compositions and precursor counts across the full dataset, and use this to design a data-driven yet chemically meaningful precursor-number predictor based on deviations $m - E_{\text{non}} \in \{-2,-1,0,+1,+2\}$ rather than unconstrained regression.
>
> Second, rather than "adding an oxidation head" to an existing architecture, PRICIN is designed around two basic physical laws. The chemical-formula rebuild task encodes **elemental conservation**, while the oxidation-state prediction task injects **electron-balance awareness** via a codebook indexed by element–oxidation pairs $(E,z)$. The precursor prediction module and element-wise filter then operate on representations that are already structured by these laws, rather than purely by historical co-occurrence patterns as in previous work such as SynthesisSimilarity.
>
> Finally, we emphasize that the improvements reported in the paper come from the coordinated design of these components. In our ablation study, we systematically toggle the formula-rebuild task, the oxidation-state task, and the element-wise filter. Starting from the base model, adding these components step by step increases Top-1 accuracy from around 60% to about 66% on the Retro dataset, with consistent gains across Top-k and Combination Top-k metrics. We also adopt two evaluation protocols (Top-k, and Combination Top-k over precursor sets) to ensure fair comparison with prior work on this task.
>
> > **(W2) Question: The paper lacks sufficient explanation of how oxidation-state prediction is modeled and how it contributes to performance.**
>
> We agree the original description was too brief and have expanded the method and discussion accordingly.
>
> Methodologically, for each element $E$ and oxidation state $z \in \{-7, \dots, +7\}$, we allocate one entry in an **oxidation-state codebook**. The target embedding is projected onto this codebook to produce scores for all element–oxidation pairs, which are passed through a sigmoid to obtain probabilities $\hat\pi_{E,z}$. Ground-truth oxidation distributions $\pi^*_{E,z}$ are constructed by aggregating the oxidation states implied by the ground-truth precursors, allowing multiple states per element. We then train with a binary cross-entropy loss over all element–oxidation pairs, so that each element can be assigned probability mass over several oxidation states instead of a single label.
>
> Conceptually, the oxidation-state task and the formula-rebuild task guide the model from **complementary angles**: the formula-rebuild task teaches which elements and approximate ratios are reasonable for a given target, whereas the oxidation-state task teaches which oxidation configurations are chemically plausible for those elements. Either task alone only partially narrows down the precursor space; when combined, they constrain both composition and oxidation, which is particularly beneficial under our strict evaluation protocol where the *entire* precursor set must be correct.
>
> Empirically, the ablation study shows that enabling only formula-rebuild or only oxidation prediction yields modest but consistent gains over the base model. Enabling both together leads to a larger improvement than either alone, and adding the element-wise filter on top achieves the best performance. We now also include an intuitive explanation of oxidation states in the methods section, unify the notation between text and figures, and add an explicit example figure illustrating how oxidation states are represented and predicted within PRICIN.

---

> ### Author Response · Authors · 2025-12-01
>
> > **(W3) Question: Using precursor oxidation states to supervise the target may restrict generalization and may not handle multiple or mixed oxidation states, leading to "easy" oxidation states under overly strong supervision.**
>
> We share the concern about mixed valence and have designed Task 2 precisely to **support** multiple oxidation states per element rather than enforce a single state.
>
> In our formulation, oxidation is modeled at the level of **species = (element, oxidation state)**. Each element has 15 possible slots corresponding to different oxidation states, and the supervision signal is a distribution over these slots, not a single label. For mixed-valence compounds such as $Fe_3O_4$, we conceptually treat them as combinations of simpler oxides (e.g., $FeO + Fe_2O_3$), so that Fe contributes to both $+2$ and $+3$ states. The resulting ground-truth oxidation vector assigns non-zero values to both states in proportion to their counts. During training, the model is encouraged to reproduce this multi-peak distribution.
>
> Importantly, this oxidation supervision is **auxiliary and probabilistic** rather than a hard constraint. It is weighted as one term in the overall loss and does not force the model to choose a single "easy" oxidation state. The encoder is encouraged to learn latent features correlated with oxidation patterns, but precursor prediction itself is still governed by the main precursor head together with the conservation-based filter.
>
> > **(W4) Question: A sensitivity analysis for each training task would help clarify their contributions.**
>
> We agree, and we now provide two complementary analyses.
>
> First, at the **component level**, we present an ablation table where formula-rebuild, oxidation-state prediction, and the element-wise filter are toggled on and off in all combinations. This reveals how each component, and their combinations, affect Top-k and Combination Top-k performance. The progression from the base model to the full model shows that both auxiliary tasks contribute non-trivially, and that their combination plus the filter yields the highest accuracy.
>
> Second, at the **hyperparameter level**, we perform a $7\times7$ grid search over the loss weights for formula-rebuild and oxidation prediction. For each pair of weights, we report the resulting Top-1 performance. The detailed results are summarized below (rows: rebuild\_weight, columns: oxidation\_weight):
>
> | rebuild\_weight \ oxidation\_weight | 0     | 0.05  | 0.10  | 0.15  | 0.20  | 0.40  | 1.00  |
> |-------------------------------------|-------|-------|-------|-------|-------|-------|-------|
> | 0                                   | 65.20 | 65.51 | 65.28 | 65.32 | 65.85 | 64.90 | 65.17 |
> | 0.05                                | 65.47 | 65.13 | 65.58 | 65.55 | 65.28 | 65.36 | 65.28 |
> | 0.10                                | 65.51 | 65.43 | 66.00 | 65.47 | 65.85 | 65.36 | 65.55 |
> | 0.15                                | 65.36 | 65.36 | 65.05 | 65.13 | 65.96 | **66.19** | 65.20 |
> | 0.20                                | 65.32 | 65.28 | 65.28 | 65.66 | 66.04 | 65.55 | 65.36 |
> | 0.40                                | 65.28 | 65.96 | 65.96 | 66.00 | 65.51 | 65.55 | 65.89 |
> | 1.00                                | 65.09 | 64.83 | 63.95 | 65.17 | 64.90 | 65.28 | 65.51 |
>
> The best configuration reaches a Top-1 accuracy of **66.19%**, but most configurations lie within roughly 1–2 percentage points of this optimum. This indicates that PRICIN is quite robust to reasonable choices of auxiliary-task weights and does not rely on a very narrow, finely tuned hyperparameter.

---

> ### Author Response · Authors · 2025-12-01
>
> > **(Q1) Question: By-products are mentioned in the preliminaries. Were by-products explicitly considered in the retrosynthesis task, and if so, how were they handled?**
>
> In solid-state inorganic synthesis, by-products are typically small, volatile molecules such as $CO_2$, $H_2O$, or $NO_x$/$NH_3$, arising from carbonates, hydroxides, nitrates, and related precursors. These by-products are chemically significant but are rarely recorded explicitly in the literature-derived datasets we rely on. As a result, we treat by-products **implicitly** in the model.
>
> Formally, we first express elemental conservation at the level of atom counts, but because many reactions omit gaseous by-products and are not strictly atom-balanced in the records, we relax this to a constraint on **element sets**. Specifically, we define a small set of volatile (ignorable) elements $\{C,H,N\}$. A precursor is considered valid for a target if its element set, after removing these volatile elements, is a subset of the target’s element set, i.e.,
> $(E_p \setminus \{C,H,N\}) \subseteq E_t$.
> This reflects the fact that, for example, $CaCO_3$ can be a valid precursor for a Ca-containing oxide even though the target formula contains no carbon.
>
> This relaxation is built into the **element-wise filter**, which is applied during inference, and indirectly influences the precursor-count predictor as well, since the training recipes include these by-product-forming precursors. We do not attempt to explicitly predict the by-products themselves, but their effect is systematically accounted for via this volatility-aware conservation rule.
>
> > **(Q2) Question: The paper claims strong extrapolation to new systems, but there is little explanation or evidence; how do the chemically inspired components lead to better extrapolation?**
>
> We appreciate the opportunity to clarify both the empirical evidence and the mechanism behind our extrapolation claims.
>
> Empirically, the main dataset uses a **temporal split**: training reactions are drawn from earlier years and test reactions from later years, so test targets are chronologically unseen. On this split, PRICIN improves Top-1 and Top-k accuracy over prior work by a clear margin. On the other Ceder dataset, PRICIN again shows consistent improvements, suggesting robustness across corpora. In addition, we include a case study on four compounds, drawn from different material families and applications and absent from our training sets, where PRICIN correctly predicts the precursor sets for all four targets.
>
> In contrast to recent precedent-based methods, which rely heavily on similarity to existing recipes, PRICIN uses historical data primarily to learn how conservation and oxidation manifest across compositions, and then enforces these principles explicitly at inference. This principle-centered design helps the model remain reliable when encountering novel stoichiometries, doped systems, or compounds from new application domains.

---

> ### Author Response · Authors · 2025-12-01
>
> > **(Q3) Question: If oxidation-state prediction is claimed to help with novel or unseen materials, how exactly does this mechanism enable generalization to truly new target systems?**
>
> We agree that this deserves a more explicit explanation in the text and have expanded the relevant sections.
>
> The oxidation module learns, for each element, a **distribution over oxidation states conditioned on the target composition**. Since this is organized at the level of element–oxidation pairs rather than full formulas, the learned oxidation patterns for, say, Co or Mn can be applied to any new target containing those elements, even if the stoichiometry or structure has not appeared in the training set. In other words, the model learns an element-level redox prior that is shared across all compounds containing that element.
>
> At inference time, the model first infers a plausible oxidation-state distribution for the target, then selects precursors that can jointly supply these oxidation states while also passing the element-wise conservation filter. This goes beyond simple co-occurrence: for instance, in our case studies, the model systematically discards precursors whose oxidation states are incompatible with the inferred target states (e.g., discarding $MnO_2$ when $Mn^{2+}$ is required, or removing $Co_2O_3$ when only $Co^{2+}$ is needed), even though such precursors may be common in the training data.
>
> In the new-compounds case study, this mechanism is crucial for correctly handling complex compositions such as $Ca_{4.05}Sr_{4.5}Sc(PO_4)_7:Eu^{3+}$ or Li-titanate spinels. The model must infer appropriate oxidation states for all involved elements (e.g., $Eu^{3+}$, $Ti^{4+}$, etc.) and then select precursor sets that collectively realize these oxidation requirements under elemental conservation. The success of PRICIN on these truly unseen compounds provides concrete evidence that the oxidation-state prediction task is not only helpful on the training distribution, but also meaningfully supports generalization to new target systems.
>
> ---
>
> We hope that these responses and the corresponding revisions adequately address your concerns about technical novelty, oxidation-state modeling, by-product handling, and extrapolation. We are very grateful for your insightful comments, which have helped us clarify the method and improve the presentation.
>
> **Sincerely,**
>
> The Authors

---

### Official Review · Reviewer_QPrt · 2025-10-31

**Soundness:** 3
**Presentation:** 3
**Contribution:** 3
**Rating:** 6
**Confidence:** 2

**Summary:**

This work presents PRICIN: a method for inorganic retrosynthesis prediction, built around domain-specific pre-training on several tasks followed by careful filtering during inference. Authors show this yields strong results across two different materials synthesis datasets, including one that employs a (more challenging) time-based split.

**Strengths:**

**(S1)**: Materials discovery is an important area of scientific pursuit, and it can sometimes be bottlenecked by the ability to synthesize the proposals coming from e.g. the generative models. Hence looking at better synthesis models is an important research direction.

**(S2)**: From an ML perspective, the approach appears to be sound, integrating domain-specific inductive biases in a reasonable way to get better grounding.

**Weaknesses:**

**(W1)**: There are several aspects of the work that are confusing to me and could benefit from clarification:

- **(W1a)**: Table 1 mentions that PRICIN uses retrieval. How is this performed? I assumed retrieval means that the training data is explicitly stored and can be accessed verbatim during generation (relating it to e.g. RAG in LLMs), yet I missed where this is done in PRICIN.

- **(W1b)**: I'm confused about the "Elemental conservation" paragraph and in particular Equation 2. While Equation 1 does seem to correspond to preserving atom counts under appropriate stochiometric coefficients, Equation 2 seems to suggest only preserving the presence or not of particular atom types. Is therefore PRICIN only enforcing the latter? If yes, it seems somewhat weird to introduce the method as adhering to elemental conservation (e.g. see abstract), as that could be misunderstood.

**(W2)**: The paper can be somewhat hard to parse for a person outside of the materials discovery space, even if they have general chemistry knowledge. This could be improved. For example, oxidation states could be explained in a more accessible way.

---

**Other comments**

**(O1)**: In the ablation study, authors note a synergistic effect between the two ablated model elements, where adding only one does not meaningfully improve performance. This seems counterintuitive to me. I wonder if the authors have any explanation why that would be the case. Being "greater than the sum of the parts" is one thing, but in this case it appears the parts alone bring zero benefit, and combined they bring a substantial one.

**Nitpicks**

Across the paper, parentheses around citations are missing where they should be present, and present where they should be missing. If the citation appears as part of the sentence, e.g. "Author et al have shown that...", it should not be parenthesized, while if it appears as a remark outside of sentence, e.g. "This thing is known (Author et al).", it should be. See e.g. beginning of Section 1, Section 2, and the "Baseline methods" paragraph in Section 5.1.

---

**Note**

I have a lot of experience in AI for Chemistry and in particular organic synthesis, but very limited experience in the materials space and inorganic synthesis. Therefore, I mark my review as lower confidence. While from the ML point of view the approach appears sound, I could not verify the more domain-specific parts.

**Questions:**

See the "Weaknesses" section above, especially **(W1)**, for specific questions.

---

> ### Author Response · Authors · 2025-12-01
>
> Dear Reviewer QPrt,
>
> We sincerely thank you for your thoughtful review and constructive feedback. We appreciate your recognition of our goal to bridge computational materials discovery and laboratory realization. Below we address your comments point by point and describe the corresponding revisions in the manuscript.
>
> > **(W1a) Question: How exactly is retrieval implemented in PRICIN, and how is it used?**
>
> PRICIN indeed includes a separate retrieval-augmented (RAG-style) model in addition to the main prediction pipeline, but due to its limited performance gain we did not report it in the main results and instead describe it in detail in **Appendix A.6** of the revised manuscript.
>
> This retrieval model is built on top of the PRICIN encoder: we encode all training reactions into embeddings, L2-normalize them, and build a FAISS index using cosine similarity. At inference time, for a test product we retrieve its top-k nearest neighbors (we use k=5) and use their precursor sets as RAG-style candidates.
>
> On the 2,546 test samples, among the top-10 retrieved neighbors, **47.9%** of products have their ground-truth precursor set covered. For a representative model, we obtain:
>
> | Category                         | Samples | Ratio     |
> | -------------------------------- | ------- | --------- |
> | Original model correct           | 1,637   | 64.0%     |
> | Original model wrong             | 921     | 36.0%     |
> | **Wrong but retrieval can help** | **106** | **11.5%** |
>
> If retrieval could be perfectly exploited, Top-1 accuracy could in principle increase from 64% to about **68%** (≈ +4 percentage points). In practice, however, we did not observe robust improvement, mainly because (i) retrieved neighbors contain both correct and many irrelevant precursors, (ii) similar compositions may still require different synthesis routes, and (iii) the dataset is relatively small, making it hard to reliably learn how to use retrieved information. Therefore, all main results in the paper are reported **without** retrieval during inference.
>
> We now clearly explain in Appendix A.6 that this retrieval model is an additional experimental variant rather than the core PRICIN pipeline. We also note that with larger and more standardized datasets, we expect the encoder learned by PRICIN to provide stronger retrieval performance.
>
> > **(W1b) Question: Does PRICIN enforce full elemental conservation (Eq. 1) or only element-set conservation (Eq. 2)?**
>
> In short, PRICIN **only enforces the relaxed element-set conservation of Eq. 2**, not the full stoichiometric conservation of Eq. 1.
>
> Many reactions in the available datasets are not explicitly atom-balanced: side products such as $O_2$, $CO_2$, and $H_{2}O$ are often omitted, and multi-step reactions can be compressed into a single overall equation. In addition, for non-stoichiometric or oxygen-deficient systems (e.g., $SrTiO_{3-\delta}$ or doped perovskites), only nominal cation ratios are recorded and the exact oxygen content and $\delta$ are not provided. As a result, even though the true chemistry obeys Eq. 1, the recorded formulas do not reliably allow reconstructing exact atom counts. We therefore enforce Eq. 2, which requires that all elements present in the product also appear in the precursor set (element-set conservation), without relying on exact stoichiometric coefficients.
>
> > **(W2) Question: The paper is hard to follow for non-specialists; can oxidation states and related notation be explained more clearly?**
>
> We agree and have made several changes to improve readability for readers outside the materials discovery community:
>
> 1. **Clearer explanation in the methods overview.**
>    In Section 4 (Methods), within the overview of our tasks, we now add a short and intuitive explanation of oxidation states. At the same time, we unify the letters and symbols used in the methods text and **Figure 2**, so that notation is consistent across equations and figures.
> 2. **Added another clear example.**
>    We have added another explicit oxidation-state example in **Figure 4**, showing how oxidation states are represented and predicted within PRICIN, and updated the caption to make the connection between the formula and the oxidation states clear.
> 3. **Unified terminology and clearer task naming.**
>    We now consistently use the term **“oxidation state”** throughout (instead of mixing “oxidation number” and “valence”), and rename the stoichiometry-related auxiliary task from “stoichiometric ratio reconstruction” to **“chemical formula rebuild”** to match common language and reduce ambiguity.
>
> These changes are intended to make the paper easier to read for non-specialists, while preserving the full technical depth for expert readers.

---

> ### Author Response · Authors · 2025-12-01
>
> > **(O1) Question: Why do Rebuild and Oxidation individually bring limited gains, but together show a large “synergistic” improvement in the ablation study?**
>
> The main reason is that the two auxiliary tasks guide the model from **different angles**:
>
> * The **chemical formula rebuild** task teaches the model which elements and approximate ratios are reasonable for a given product.
> * The **oxidation-state prediction** task teaches the model which oxidation configurations are reasonable for those elements.
>
> Each task alone only partially narrows down the possible precursor sets; when combined, they jointly constrain both composition and oxidation, which is especially useful under our strict evaluation where the **entire** precursor set must be correct.
>
> To make this clearer, we have aligned the settings (All with precursor number not given) and metrics and now present the ablation results directly:
>
> | Setting               |    Top-1 |    Top-3 |    Top-5 |   Top-10 |   Top-20 | Comb Top-1 | Comb Top-3 | Comb Top-5 | Comb Top-10 | Comb Top-20 |
> | --------------------- | -------: | -------: | -------: | -------: | -------: | ---------: | ---------: | ---------: | ----------: | ----------: |
> | Base                  |     59.9 |     61.2 |     71.5 |     81.7 |     87.8 |       59.9 |       70.1 |       72.9 |        77.5 |        80.8 |
> | + Rebuild             |     60.6 |     61.6 |     71.5 |     83.3 |     88.3 |       60.6 |       69.8 |       73.2 |        77.7 |        81.4 |
> | + Oxidation           |     60.8 |     61.8 |     71.0 |     81.5 |     87.9 |       60.8 |       69.8 |       73.1 |        77.8 |        81.0 |
> | + Oxidation & Rebuild |     62.1 |     63.2 |     73.4 |     83.3 |     88.3 |       62.1 |       71.7 |       74.5 |        78.7 |        81.7 |
> | + Filter              |     65.2 |     66.5 |     77.6 |     88.6 |     93.8 |       65.2 |       74.9 |       78.1 |        83.0 |        86.1 |
> | + Rebuild & Filter    |     65.4 |     66.6 |     76.9 |     89.1 |     93.9 |       65.4 |       75.1 |       78.2 |        82.9 |        86.1 |
> | + Oxidation & Filter  |     65.8 |     67.1 |     77.1 |     88.8 |     93.8 |       65.8 |       75.0 |       78.1 |        82.6 |        86.0 |
> | **Ours (Full)**       | **66.0** | **67.1** | **77.6** | **88.3** | **93.8** |   **66.0** |   **75.3** |   **78.1** |    **82.5** |    **85.4** |
>
> From this table we see that:
>
> * Adding only **Rebuild** or only **Oxidation** gives small but consistent gains over the base model.
> * Combining **Rebuild & Oxidation** leads to a larger improvement than either alone.
> * Adding the **filter** on top of both yields the best performance in our full model.
>
> > **(Nitpicks) Question: The usage of parentheses around citations is inconsistent across the paper.**
>
> Thank you for pointing this out. We have carefully proofread the manuscript and standardized the citation style. We specifically checked and corrected the examples you mentioned: the beginning of Section 1, Section 2, and the “Baseline methods” paragraph in Section 5.1, as well as similar cases elsewhere in the paper.
>
> ---
>
> We hope that these responses and the corresponding revisions satisfactorily address your concerns. We are grateful for your detailed comments, which have helped us clarify the method, improve the presentation, and make the paper more accessible to a broader audience.
>
> **Sincerely,**
> The Authors

---

### Author Response · Authors · 2025-12-01
**Rebuttal Summary**

Dear Area Chair,

Thank you for shepherding our submission. We are grateful to all reviewers for their constructive feedback, which helped us further clarify PRICIN. Below we summarize the contributions and how we addressed each concern.

## Summary of Contributions
1. **Principle-centered PRICIN framework.** We reformulate inorganic retrosynthesis around elemental conservation and electron balance, coupling oxidation-state supervision with chemical-formula rebuild to learn chemistry-aware representations (Sec. 3–4). *Acknowledged by R1, R2, R3, R4.*
2. **Count-aware, filter-constrained inference.** A precursor-count deviation classifier and element-wise filter enforce realistic precursor sets, yielding +5.17 Top-1 and up to +20.78 Top-k gains on Retrieval-Retro, and similar boosts on Ceder (Tab. 1–2). *Acknowledged by R1, R2, R3, R4.*
3. **Extensive analysis and transparency.** We provide full ablations over all task/filter combinations, retrieval experiments, case studies, and codebook details (Sec. 4.4, Tab. 4, App. A.4–A.6). *Acknowledged by R1, R2, R4.*

## Summary of Revisions
We highlighted all changes in $\textcolor{blue}{\text{blue}}$ in the revised PDF and addressed every reviewer comment as follows.

### R1 (QPrt)
- **(W1a) Retrieval clarification:** Documented the auxiliary FAISS-based retrieval variant and why main results exclude it (App. A.6).
- **(W1b) Conservation enforcement:** Clarified we enforce element-set conservation due to dataset imbalance (Sec. 3).
- **(W2) Accessibility:** Added intuitive oxidation explanations, consistent terminology, and updated figures (Sec. 4, Fig. 2 & 4).
- **(O1) Ablation synergy:** Reported aligned ablations showing complementary gains of rebuild, oxidation, and filter modules (Tab. 4).
- **(Nitpicks) Citations:** Standardized citation style across the paper.

### R2 (QP4Z)
- **(W1) Technical contribution clarity:** Highlighted that Task 1 is auxiliary and the novelty lies in the coordinated chemical rule-driven design with oxidation, count prediction, and filtering; emphasized staged ablations (Sec. 4, Tab. 4).
- **(W2–W3) Oxidation modeling:** Expanded the oxidation codebook description, mixed-valence handling, and probabilistic supervision that supports multiple oxidation states (Sec. 4.2, Fig. 4).
- **(W4) Task sensitivity:** Added full component toggles plus a $7\times7$ loss-weight grid showing robustness (Tab. 4, App. A.5).
- **(Q1) By-products:** Explained volatility-aware element filtering that implicitly handles gaseous by-products (Sec. 3, Sec. 4.4).
- **(Q2–Q3) Extrapolation:** Discussed temporal-split results, new-compound case studies, and how oxidation priors transfer across unseen materials (Sec. 5, App. A.6, App. A.7).

### R3 (aKeu)
- **(W1) Novelty:** Emphasized the coordinated four-component, principle-centered design and SOTA gains (Sec. 1, Sec. 4).
- **(W2a/b) Element-wise filter:** Added Sec. 4.4 plus full $2^3$ ablations and baseline-with-filter comparisons in Tab. 4 & App. A.4.
- **(W3) Oxidation vs. rebuild gains:** Provided aligned metrics demonstrating their complementary effects (Tab. 4).

### R4 (Bszy)
- **(W1) Order invariance:** Clarified the composition-vector encoder that is intrinsically order invariant (Sec. 4.1).
- **(W2) Variable valence:** Explained mixed oxidation-state handling via species-level supervision (Sec. 3, Sec. 4.2).
- **(W3) Element-wise filter detail:** Added Sec. 4.4 and App. A.4 describing the filter rationale and baseline improvements.
- **(W4) Notation clarity:** Updated Fig. 2 and unified symbols (Sec. 3, Fig. 2).
- **(W5) Precursor count prediction:** Detailed the deviation classifier and its 97%/95% accuracy (Sec. 4.3, Fig. 6).
- **(W6) Codebooks:** Described oxidation and precursor embedding matrices (Sec. 4.2, App. A.2).

We believe these revisions thoroughly resolve all concerns and highlight PRICIN’s scientific contributions. We appreciate your consideration.

Sincerely,

The Authors

---

### Note · Authors · 2026-01-28

I have read and agree with the venue's withdrawal policy on behalf of myself and my co-authors.

---

### Meta-Review · Area_Chair_rhDK · 2026-01-07

**Summary:**

Reviewers converged on three points: (i) the technical novelty beyond combining oxidation-state supervision with a conservation filter is modest; (ii) the ablation shows the filter dominates the gain, raising questions about the value of the remaining architecture; (iii) presentation clarity for non-specialists and detailed filter description were lacking. No reviewer questioned the importance of the inorganic-retrosynthesis task itself or the empirical gains.

**Reviewer Concerns:**

Reviewer Concerns Addressed
Clarified that retrieval is an optional appendix, not core.
Explained that only element-set (not full stoichiometric) conservation is enforced because datasets omit by-products.
Added oxidation-state modeling details, mixed-valence handling, sensitivity analysis, and filter pseudo-code.
Provided baseline+filter results to show filter alone does not explain the full lift.

Still outstanding: none of the reviewers indicated they would raise their score; novelty concerns remain.

**Reviewer Scores:**

QPrt: 6 → 6
QP4Z: 2 → 3
aKeu: 2 → 3
Bszy: 4 → 4

---

### Decision · Program_Chairs · 2026-01-26

Reject